# Blood transcriptome based biomarkers for human circadian phase

**Emma E Laing[1]\*[†], Carla S Möller-Levet[2][†], Norman Poh[3][†], Nayantara Santhi[4], Simon N Archer[4], Derk-Jan Dijk[4]\***

[1]Department of Microbial Sciences, School of Biosciences and Medicine, Faculty of Health and Medical Sciences, University of Surrey, Guildford, United Kingdom; [2]Bioinformatics Core Facility, Faculty of Health and Medical Sciences, University of Surrey, Guildford, United Kingdom; [3]Department of Computer Science, Faculty of Engineering and Physical Sciences, University of Surrey, Guildford, United Kingdom; [4]Surrey Sleep Research Centre, School of Biosciences and Medicine, Faculty of Health and Medical Sciences, University of Surrey, Guildford, United Kingdom

**Abstract** Diagnosis and treatment of circadian rhythm sleep-wake disorders both require assessment of circadian phase of the brain's circadian pacemaker. The gold-standard univariate method is based on collection of a 24-hr time series of plasma melatonin, a suprachiasmatic nucleus-driven pineal hormone. We developed and validated a multivariate whole-blood mRNA-based predictor of melatonin phase which requires few samples. Transcriptome data were collected under normal, sleep-deprivation and abnormal sleep-timing conditions to assess robustness of the predictor. Partial least square regression (PLSR), applied to the transcriptome, identified a set of 100 biomarkers primarily related to glucocorticoid signaling and immune function. Validation showed that PLSR-based predictors outperform published blood-derived circadian phase predictors. When given one sample as input, the $R^2$ of predicted vs observed phase was 0.74, whereas for two samples taken 12 hr apart, $R^2$ was 0.90. This blood transcriptome-based model enables assessment of circadian phase from a few samples.

\*For correspondence: e.laing@ surrey.ac.uk (EEL); d.j.dijk@surrey. ac.uk (D-JD)

[†]These authors contributed equally to this work

**Competing interests:** The authors declare that no competing interests exist.

## Introduction

Many physiological and molecular variables exhibit 24 hr rhythmicity generated by central and peripheral circadian oscillators, as well as environmental and behavioral influences (*Morris et al., 2012*; *Mohawk et al., 2012*). The master circadian pacemaker located in the suprachiasmatic nucleus (SCN) of the hypothalamus drives 24 hr rhythmicity in endocrine and neural factors such as cortisol, melatonin and the autonomic nervous system (*Brancaccio et al., 2014*; *Kalsbeek and Fliers, 2013*). Under normal conditions, the SCN is synchronized to the 24 hr light-dark (L-D) cycle by a light-input pathway (*Lucas et al., 2014*). The effects of light on the SCN depend on the timing, that is, the circadian phase of the SCN, at which light is administered (*St Hilaire et al., 2012*). The SCN is also sensitive to feedback from the pineal melatonin rhythm which impinges on the SCN through its melatonin receptors (*Pevet and Challet, 2011*). Exogenous melatonin can phase-shift and entrain the SCN, but its effects depend on the circadian phase of melatonin administration (*Lewy et al., 2005*).

Synchronization of the SCN to the L-D cycle and synchronization of the sleep-wake cycle to the SCN are disrupted during shift work and jet lag and in circadian rhythm sleep-wake disorders (CRSWDs). Examples of CRSWDs are delayed sleep-wake phase disorder, which is prevalent in adolescents (*Micic et al., 2016*), and non-24-hr sleep-wake disorder, such as occurs in blind individuals (*Sack et al., 1992*; *Sack and Lewy, 2001*). Treatment of these circadian disorders by melatonin or

light exploits the circadian phase-dependent response of the SCN to these stimuli (*Auger et al., 2015*). Optimal treatment requires that the circadian phase of the SCN is known and is thereby one example of chronotherapy, an increasingly important field in drug development and medicine (*Dallmann et al., 2016*). Knowledge of the circadian phase of the SCN is also informative for the diagnosis of CRSWDs, assessment of adaptation to new time zones, adaptation to shift work (*Mullington et al., 2016*) and identification of circadian rhythm disturbances in psychiatric conditions. Conventionally, circadian phase of the SCN is assessed through long-term collection of biomarkers in the field or in the clinic. Examples are repeated urinary 6-sulfatoxy melatonin assessments (*Flynn-Evans et al., 2014*; *Lockley et al., 2015*), long-term core body temperature recordings (*Klein et al., 1993*), or frequent blood or saliva sampling in dim light, for the assessment of melatonin rhythm (*Klerman et al., 2002*; *Danilenko et al., 2014*). The plasma rhythm of melatonin is driven by rhythmic melatonin synthesis in the pineal and under control of the SCN (*Arendt, 2005*). In human clinical research, melatonin is therefore considered a gold-standard proxy for SCN phase. However, circadian phase assessments based on the plasma melatonin rhythm are time consuming, burdensome and costly. This would be greatly reduced if circadian phase could be assessed from only one or few blood samples.

At the molecular level, circadian oscillators consist of a network of genes that, through a transcriptional-translational feedback loop, generates tissue-specific circadian rhythmicity in transcription and translation (*Partch et al., 2014*; *Koike et al., 2012*). In human blood, 6.4–8.8% of the transcriptome exhibits circadian rhythmicity when sleep occurs in phase with the melatonin cycle and in the absence of a sleep-wake cycle (*Archer et al., 2014*; *Möller-Levet et al., 2013*). This peripheral circadian rhythmicity is, however, markedly disrupted when sleep occurs out of phase with the melatonin rhythm. Under these conditions, only 1% of transcripts are rhythmic (*Archer et al., 2014*). The human blood transcriptome is also affected by both chronic sleep restriction and acute total sleep deprivation (*Möller-Levet et al., 2013*). This sensitivity of the peripheral circadian transcriptome to factors such as sleep timing and sleep restriction poses a significant challenge for the assessment of circadian phase from the blood transcriptome. In many target conditions and patient populations, for example, shift work and non-24-hr sleep-wake disorder, sleep timing and duration are altered. Nevertheless, one or two blood transcriptome mRNA abundance profiles could potentially contain information about the circadian phase at which these samples were taken under all these conditions.

Phase prediction methods based on omics data from a few samples of blood or other tissues have been developed. Some methods selected the transcripts (features) to be used as predictors based on existing knowledge of their rhythmic expression and/or involvement in circadian processes (*Lech et al., 2016*), that is, a transcriptome wide search was not performed. Other methods required predictors to be rhythmic. These methods first fitted a rhythmic function (e.g. cosine-wave [*Ueda et al., 2004*; *Minami et al., 2009*] or rhythmic spline [*Hughey et al., 2016*]) to time-series of transcriptomic or metabolomic data to identify (24 hr) rhythmically expressed features, which were then used as predictors. None of these methods, either applied to the transcriptomic profiles of different mouse organs (*Ueda et al., 2004*; *Minami et al., 2009*), or the metabolomic (*Minami et al., 2009*) or transcriptomic profiles (*Lech et al., 2016*) of human blood, specifically aimed to predict the phase of the melatonin rhythm. Moreover, the performance of these methods was not assessed in conditions of desynchrony of the sleep-wake cycle and the melatonin rhythm.

Here, our aims are (1) to develop and validate a circadian phase prediction method by a transcriptome wide search for a set of features that is capable of predicting the circadian phase of the plasma melatonin rhythm, regardless of sleep history and timing, using a few blood samples; (2) to compare the performance of this method to published approaches; and (3) identify the processes associated with the set of identified biomarkers.

## Results

### Overview of development and validation of models for predicting circadian phase

mRNA abundance features and melatonin concentrations were assessed in blood samples collected under four sleep timing conditions (*Archer et al., 2014*; *Möller-Levet et al., 2013*) (*Figure 1d,h*): (i) Sleep in phase with melatonin; (ii) Sleep out of phase with melatonin; (iii) total sleep deprivation, no

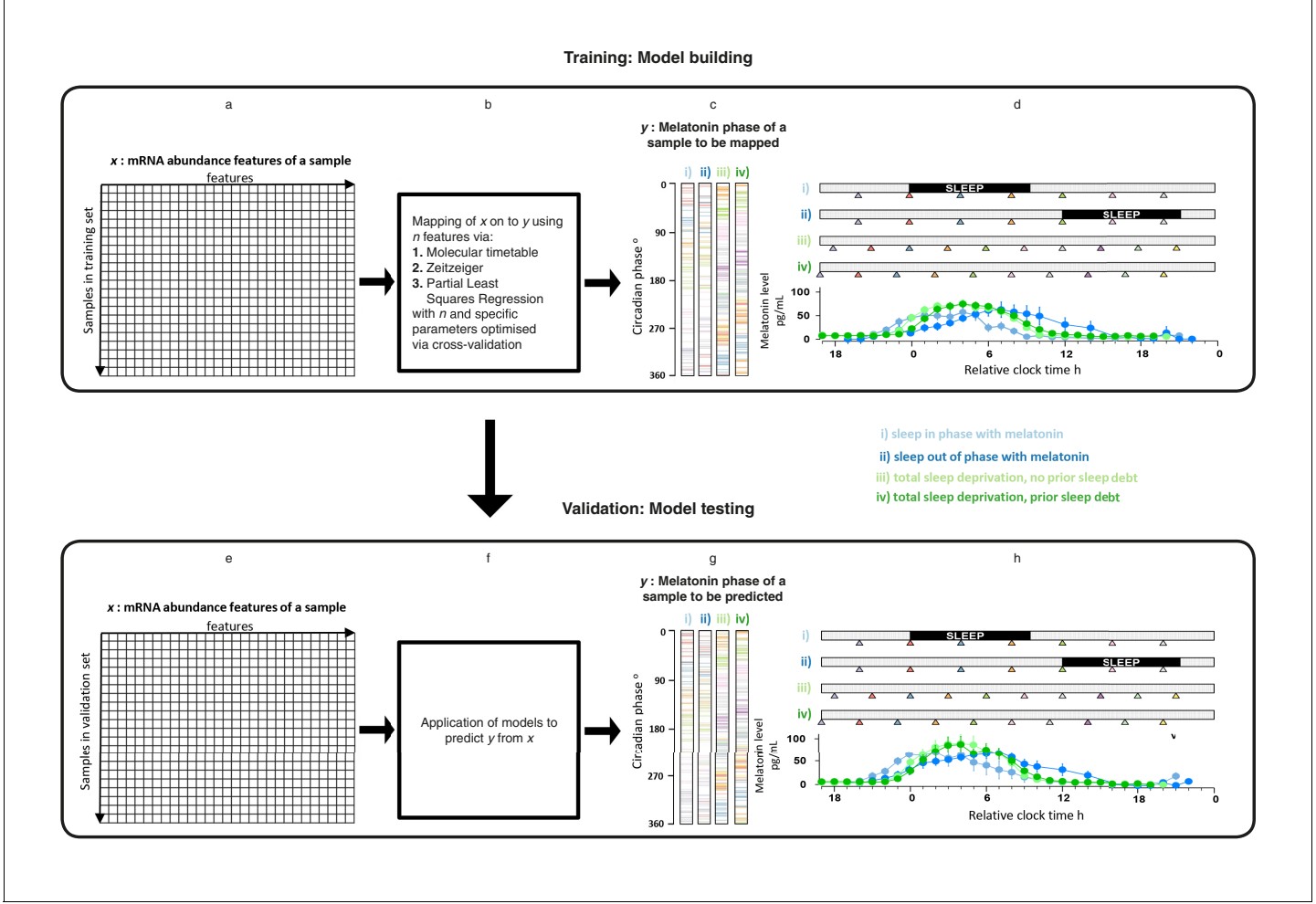

**Figure 1.** Development and validation of models for predicting melatonin phase from transcriptomic samples. mRNA abundance and melatonin data from 53 participants collected in four conditions (panels **c, d, g and h**) were partitioned into two groups: a training set (329 mRNA samples from 26 participants) to build the model and a validation set (349 mRNA samples from 27 participant). For each participant in each condition, hourly melatonin samples (closed circles in plots represent the average melatonin profile of participant in a given experimental condition) and multiple transcriptomic samples (colored triangles) separated by 3 (conditions i and ii) or 4 hr (conditions ii and iv) were collected. Based on a participants' melatonin profile, each transcriptomic sample for that participant was assigned a circadian phase and plotted separately for each of the conditions (panels **c, g**). The coloring of circadian bars in panels **c** and **g** correspond to the coloring of the transcriptomic sampling time-point triangles in panels **d, h**. For example, the third mRNA sample (blue triangle) in condition iv was on average taken just before the onset of melatonin (dark green melatonin profile) and the blue horizontal bars in condition iv panel of panel **c** therefore are located around 300 degrees. This shows the distribution of circadian phases obtained at each sampling point across participants in the training and validation set. These data were used to construct and evaluate the Molecular timetable, Zeitzeiger and Partial Least Square Regression (PLSR) based models. We mapped the melatonin phase *y* (panels **c, g**) onto the transcriptomic profile comprising *x* features in the training set (panels **a, e**). Parameters to build the model were selected by performing leave-one-participant-out cross-validation of the training set (panel **b**). The final trained model was applied to the validation set for assessing the overall performance of the model (panel **f**).

prior sleep debt and (iv) total sleep deprivation, with prior sleep debt. The melatonin rhythm persists in the absence of sleep and when sleep occurs during the daytime provided that light levels are low, as was the case in these experiments (*Figure 1d,h*). Each of 678 genome-wide mRNA abundance profiles were assigned a circadian melatonin phase by referencing it to the onset of melatonin secretion (0 degrees) in the corresponding individual time series (*Archer et al., 2014*; *Möller-Levet et al., 2013*). Melatonin phases of mRNA samples spanned the entire circadian cycle (*Figure 1c,g*). To develop practical models, we divided our data set into a training set comprising 329 samples and a validation set comprising 349 samples (see Materials and methods).

## Rhythmicity of the blood transcriptome across conditions

Existing methods, such as the molecular timetable method (*Ueda et al., 2004*) rely on identifying features that correlate well with a 24 hr cosine wave. We estimated rhythmicity of the transcriptome by computing the correlation between a cosine wave and the time series of all features in each of our four data sets and across all four data sets separately for the training and validation set (see Materials and methods). Plotting the $r^2$ value against the rank of the correlation shows that overall 24 hr rhythmicity in the transcriptome is most prominent in the 'sleep in phase with melatonin' condition and smallest in the 'sleep out of phase with melatonin' condition (*Figure 2a,d*).

## Molecular timetable and Zeitzeiger methods

To test the performance of the timetable method (*Ueda et al., 2004*), we identified features that were robustly rhythmic across all conditions in the training set. Seventy-three features satisfied the criterion of a minimum correlation of 0.3 in each condition and across all four conditions (see Materials and methods and *Figure 2—figure supplement 1* for threshold derivation; *Figure 2—figure supplement 2* for temporal profiles of genes comprising the timetable model derived from our data). The 'rhythmicity' of these 73 features was compared to the 'rhythmicity' of previously published phase marker gene sets of (*Lech et al., 2016*; *Hughey et al., 2016*) by rank ordering their correlation coefficients. The 'rhythmicity' of the 73 identified features is in general, more robust than the rhythmicity of the genes used in previous approaches (*Figure 2b,e*). In particular, it is the condition 'sleeping out of phase with melatonin' (condition ii in *Figure 1*) (*Figure 2b,c,e,f*) that appears to exclude many of the previously published phase marker genes as good candidates for features to be included in a timetable model (see *Figure 2—figure supplement 3*, *Figure 2—figure supplement 4*, for temporal profiles of genes in the previously reported phase marker gene sets; *Supplementary file 1A* for their corresponding correlation $r$ coefficients). Indeed, the set of 73 features does not significantly overlap with previously reported sets of phase predictors (*Supplementary file 1B* for comparison of models' gene lists). Gene ontology (GO) functional enrichment analysis of the 73 features showed that they were significantly (Benjamini and Hochberg adjusted p-value<0.05) associated with the following biological processes: T cell receptor V(D)J recombination, somatic diversification of T cell receptor genes and somatic recombination of T cell receptor gene segments (*Supplementary file 2*).

The performance of our one-sample timetable model based on the 73 identified features is summarized and compared to other models in *Table 1*, *Figure 3* and *Figure 3—figure supplement 1*. The timetable model using these 73 features identified in the training set predicts the circadian phase of samples in the validation set on average within 9 min of the observed melatonin phase which is comparable to the 15 min and −2 min bias when the original gene lists of (*Lech et al., 2016*) and (*Hughey et al., 2016*) were used, respectively (*Table 1*, *Figure 3c,f,i*). Any identified systematic error in a model can be corrected for when optimizing a final model for future applications and the relevant measure of performance is the standard deviation of the error. The molecular timetable model comprising the 73 features we identified had the least variation of prediction error. The errors of the 73-feature model varied across the circadian cycle (*Table 1*, *Figure 3*, *Figure 3—figure supplement 1b,e,h*). Our one-sample molecular timetable model outperformed the models constructed from published gene sets of *Lech et al. (2016)* and *Hughey et al. (2016)*, when considering the 'proportion of samples with ≤2 hr error' (*Table 1*, *Figure 3j*) and the $R^2$ (*Table 1*, *Figure 3—figure supplement 1a,d,g*). Nevertheless, it is clear that each of these one-sample molecular timetable models perform poorly when applied to our human transcriptome data.

We next used 'Zeitzeiger' (*Hughey et al., 2016*), which can be considered a machine learning approach to the molecular timetable method, to identify circadian phase biomarkers in our data sets. In Zeitzeiger, a high-dimensional transcriptomic profile sample is first projected into a low-dimensional space using the maximum likelihood principle before assigning the circadian time to that sample. Our implementation of Zeitzeiger identified 107 features as the optimal number with which to predict the circadian phase of one sample (see Materials and methods and *Figure 3—figure supplement 2*). Only three genes, *PER1*, *NR1D1* and *NR1D2*, from the original Zeitzeiger gene list identified in animal tissues (*Hughey et al., 2016*) were also identified as predictors when applied to our human blood transcriptome data. Of the 107 features we identified using Zeitzeiger, 23 were in common with the molecular timetable method (see *Supplementary file 1B*). The 107 features

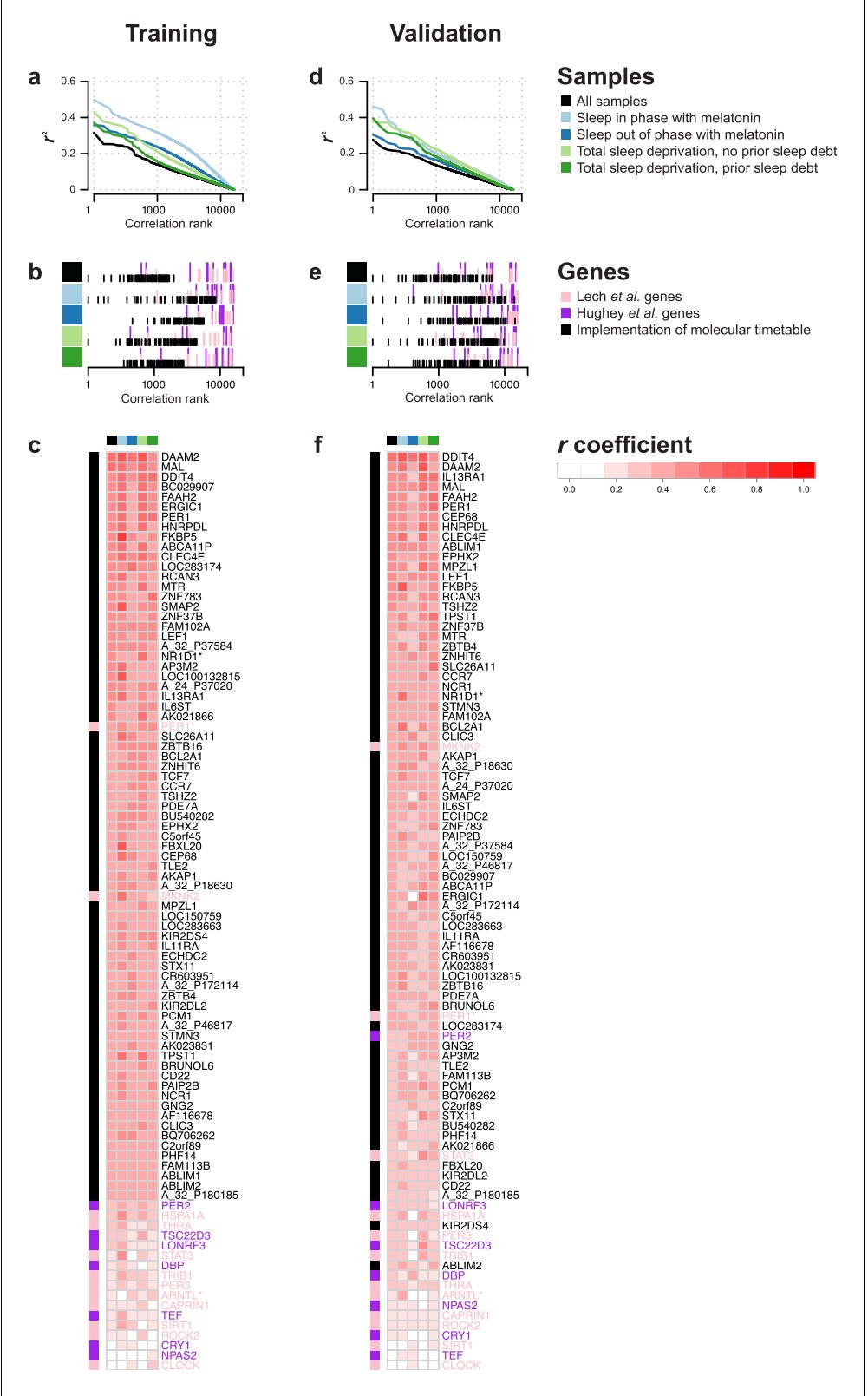

**Figure 2.** Rhythmicity in the conditions and identification of molecular timetable genes. (**a**, **d**) Square of correlation value ($r^2$) vs. rank of correlation as a measure of overall 24 hr rhythmicity in the transcriptome, separately and across conditions. For each transcript, the correlation with a cosine wave was computed, squared and the entire transcriptome was then rank-ordered separately for each conditions and across conditions. (**b**, **e**) Rank of the correlation of phase marker genes from different phase marker lists across conditions (indicated by color panel to the left) for participants in the

*Figure 2 continued on next page*

*Figure 2 continued*

training (**b**) and validation set (**e**). First, correlations of all ~26K features were ranked separately for conditions and training and validation set and then the rank of identified phase marker genes was identified and plotted. (**c**,**f**) Heatmap of *r* values of correlation between a 24 hr cosine wave and mRNA abundance profiles of a feature(s) targeting a gene identified as a phase marker by our molecular timetable, and the circadian phase marker genes published in *Lech et al. (2016)* and *Hughey et al. (2016)*. Correlations were computed separately for each of the four conditions (indicated by the color panel at the top of the heatmap) and all conditions combined for the training (**c**) or validation set (**f**). Rows in the heatmap are ordered based on the correlation column for the 'all conditions' dataset, that is, the column indicated with black. Data source files: *Figure 2—source data 1*.

The following source data and figure supplements are available for figure 2:

**Source data 1.** Source data file for generating panels in *Figure 2*.

**Figure supplement 1.** Identification of correlation cutoff threshold for the construction of a molecular timetable.

**Figure supplement 2.** mRNA abundance profiles for genes belonging to the phase marker gene list generated by our implementation of the molecular timetable model.

**Figure supplement 3.** mRNA abundance profiles for genes belonging to the phase marker gene list published in (*Lech et al., 2016*).

**Figure supplement 4.** mRNA abundance profiles for genes belonging to the phase marker gene list published in (*Hughey et al., 2016*).

identified by the Zeitzeiger model did not show any significant functional GO enrichment (*Supplementary file 2*). The performance of the Zeitzeiger model, when applied to our validation set, is somewhat similar to that of the molecular timetable model across all but two of the performance metrics (*Table 1*, *Figure 3a–f and j*). Both methods are similar in their approach, constructing a model based on features identified as being rhythmic within a data set, so this is perhaps

**Table 1.** Performance of trained models when used to predict the circadian phase of samples in the validation set. NS indicates not significant.

|  | Average error (minutes) | Standard Deviation of Error (hours: minutes) | Circadian variation of error (P-value of ANOVA) | Proportion of samples with $\leq$2 hr error | $R^2$ of predicted vs observed phase |
|---|---|---|---|---|---|
| Genes-from (*Lech et al., 2016*) - one sample | 15 | 5:32 | NS | 28% | 0.28 |
| Genes-from (*Hughey et al., 2016*) - one sample | −2 | 5:23 | <0.01 | 30% | 0.32 |
| Timetable - one sample | 9 | 4:38 | <0.01 | 40% | 0.49 |
| Zeitzeiger - one sample | −0.4 | 4:44 | NS | 36% | 0.47 |
| Partial Least Square Regression - one sample | −18 | 3:17 | NS | 54% | 0.74 |
| Genes-from (*Lech et al., 2016*) - two samples | 20 | 4:05 | 0.05 | 35% | 0.60 |
| Genes-from (*Hughey et al., 2016*) - two samples | −0.65 | 3:58 | 0.03 | 41% | 0.63 |
| Timtable - two samples | 11 | 3:38 | <0.01 | 43% | 0.69 |
| Zeitzeiger - two samples | -2 | 3:36 | NS | 47% | 0.69 |
| Partial Least Square Regression - two samples | −16 | 2:39 | NS | 62% | 0.83 |
| Genes-from (*Lech et al., 2016*) - three samples | 24 | 3:21 | 0.05 | 45% | 0.73 |
| Genes-from (*Hughey et al., 2016*) - three samples | 8 | 3:19 | NS | 47% | 0.74 |
| Timetable - three samples | -3 | 2:46 | <0.01 | 51% | 0.82 |
| Zeitzeiger - three samples | 4 | 3:03 | NS | 49% | 0.78 |
| Partial Least Square Regression - three samples | −11 | 2:15 | NS | 71% | 0.88 |
| Timetable - Differential two samples | −33 | 2:28 | NS | 71% | 0.78 |
| Partial Least Square Regression-Differential two samples | −18 | 1:41 | NS | 82% | 0.90 |

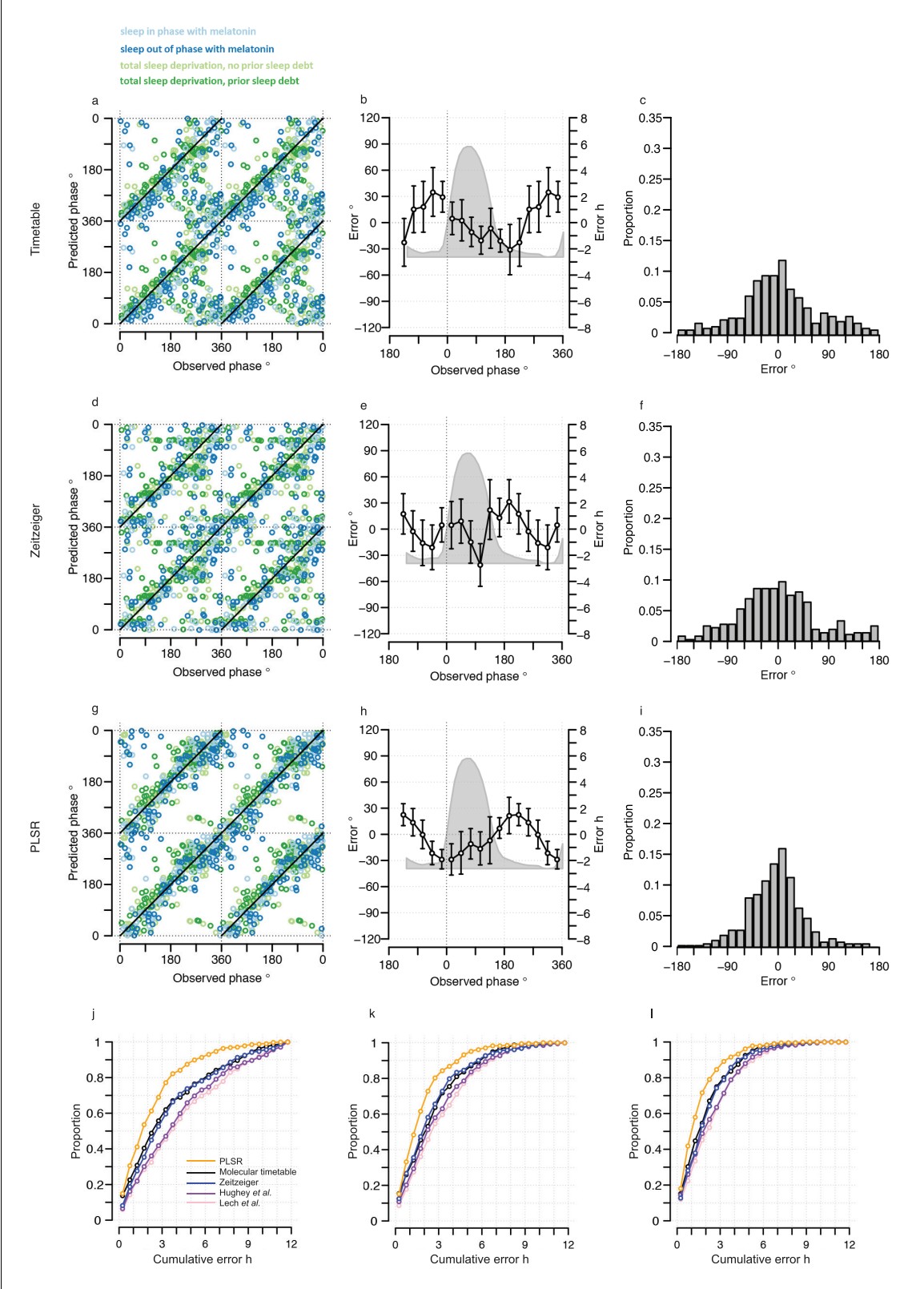

**Figure 3.** Performance of one-sample models derived from the training set when used to predict circadian phase in the validation set. (**a**, **d** and **g**) Predicted circadian phase of a blood sample vs. the observed melatonin phase for each sample in the validation set for the one-sample molecular timetable model (**a**), the one-sample Zeitzeiger model (**d**) and the one-sample PLSR-based model (**g**). Circadian phase 0 corresponds to the onset of the melatonin rhythm, which across conditions occurred on average at 23:30 ± 15 min (across all conditions and the training and validation data sets).
*Figure 3 continued on next page*

*Figure 3 continued*

Please note that the data are double-plotted. The line of unity represents perfect prediction. (**b**, **e** and **h**) Average (mean) error and 95% confidence intervals of the mean error per 30°, that is 120 min, for each 30° circadian phase bins across the circadian cycle, for each of the models. The grey area plot represents the average melatonin profile. Panels (**c**, **f** and **i**) frequency distribution of errors. Errors were sorted into 15°, that is, 60 min bins for each model. For each of the models, the mode of the distribution is near 0, but the width of the distribution varies across models. (**j**) Proportion of predictions based on one sample vs. cumulative error as a measure of overall performance. (**k**) Proportion of predictions based on two consecutive samples vs. cumulative two sample average error as a measure of overall performance. (**l**) Proportion of predictions based on three consecutive samples vs. cumulative three sample average error as a measure of overall performance. Data source file *Figure 3—source data 1*.

The following source data and figure supplements are available for figure 3:

**Source data 1.** Source data file for generating panels in *Figure 3*.

**Figure supplement 1.** Performance of one-sample molecular timetable models when used to predict the circadian phase of samples in the validation set.

**Figure supplement 2.** Parameter selection for constructing a Zeitzeiger based model.

**Figure supplement 3.** Phase profiles of genes in the phase marker lists for the molecular timetable and Zeitzeiger models.

**Figure supplement 4.** Selection of number of abundance features and latent factors and pseudo time-course of latent factor scores for Partial Least Squares Regression (PLSR)-based models.

**Figure supplement 5.** Comparison of the number of samples used as input vs, accuracy for each circadian phase prediction method.

expected. Unlike the molecular timetable, errors did not vary significantly across circadian bins. The reduced error of the Zeitzeiger model may be due to inclusion of a greater number of features, their maxima phases providing greater coverage of 360 degrees with which to more accurately predict circadian time (see *Figure 3—figure supplement 3*).

## Partial least square regression method

In view of the suboptimal performance of the approaches described above, we applied another method taking into account the following considerations. We have a much larger number of variables (each of our transcriptomic profiles comprises $N$ (~26 K) quantitative features) than available observations (melatonin phases, one per transcriptomic profile). Furthermore, the variables are not necessarily independent, for example, co-expression of transcripts and/or some features targeting the same gene (transcript). In this instance, a suitable approach to developing a predictive model is to use Partial Least Squares Regression (PLSR) (*Boulesteix and Strimmer, 2007*). In PLSR, the dimensionality of the predictor set (transcriptome profile $x$) is reduced by projecting *both* the predictor set and response variable (melatonin phase $y$) into orthogonal latent (hidden) spaces, referenced by latent factors, that are relevant for predicting the response variable, without directly prioritizing any underlying time-dependency within the data set (see Materials and methods). Factor loadings, the correlation between each feature and each factor, can then be used to select features *a posteriori* to produce the circadian phase prediction method. Features with the most weight (loading) across latent factors are of interest and/or the better predictors.

## One-sample circadian phase PLSR-based prediction model

For predictions based on one blood sample, the optimal PLSR model parameters were a combination of five latent factors and 100 mRNA abundance features (see *Figure 3—figure supplement 4a and b* and Materials and methods; see *Supplementary file 1C* for the list of features and *Supplementary file 1B* for the list of 80 corresponding genes). The representation of a given transcriptomic sample in a given latent factor (space) is provided by the latent factor score of that sample, a linear combination of feature values multiplied by the corresponding latent factor loadings. The pseudo time-course of factor scores of the five latent factors (presented in *Figure 3—figure supplement 4b*) are rhythmic in nature ($R^2$ of fit to a cosine wave range of 0.44–0.51), with phases of

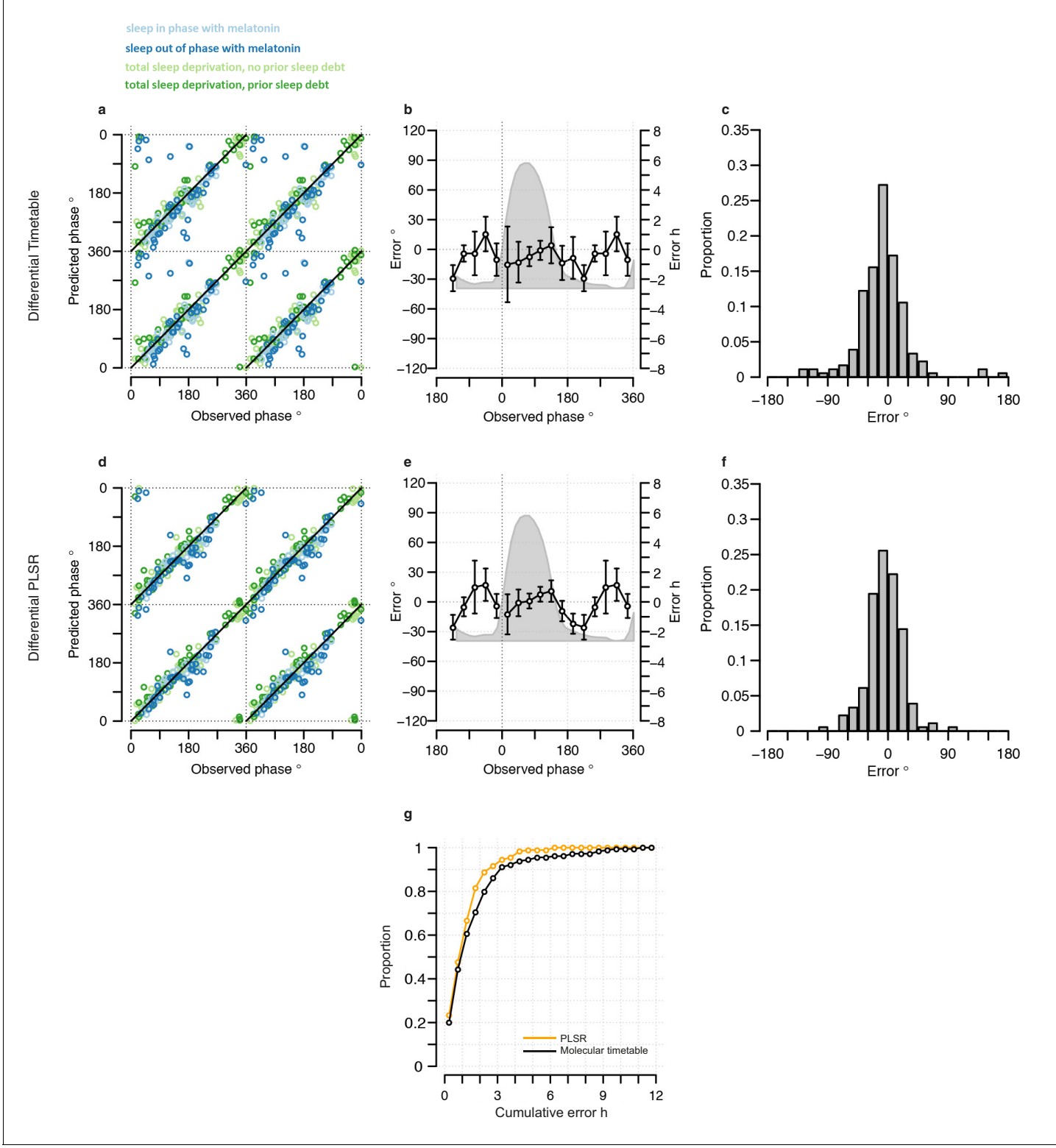

**Figure 4.** Performance of two-sample differential mRNA abundance-based models when used to predict the circadian phase in the validation set. (**a** and **d**) Predicted circadian phase of a blood sample vs. the observed circadian melatonin phase for each sample in the validation set for the two-sample differential molecular timetable model (**a**) and the two-sample differential PLSR-based model (**d**). Data are double-plotted. (**b** and **e**) Average (mean) error and 95% confidence intervals of the mean error per 30°, that is, 120 min, for each 30° circadian phase bins across the circadian cycle, for each of the models. The grey area plot represents the average melatonin profile. (**c** and **f**) Frequency distribution of errors. Errors were sorted into 15°,
*Figure 4 continued on next page*

Figure 4 continued

that is, 60 min bins, for each model. For each of the models, the mode of the distribution is near 0, but the width of the distribution varies across models. (g) Performance of the differential PLSR (orange), and the differential molecular timetable method (black) quantified by proportion of samples vs. cumulative error. Data source file *Figure 4—source data 1*.

The following source data is available for figure 4:

**Source data 1.** Source data file for generating panels in *Figure 4*.

the maxima covering 360 degrees (58°, 250°, 56°, 310° and 280°, respectively) and all have a similar amplitude (range of 0.016–0.018). Each latent factor was characterized via functional enrichment analysis of the 200 most weighted features (see *Supplementary file 2* for lists).

Latent factor 1, the factor that has the greatest covariation of *x* and *y*, gives greatest weight to features significantly enriched for biological processes linked to immune and inflammatory response (e.g. *TCF7, LEF1, NFAM1, TFE3, GIMAP5, PRKCA, GAB2, CD28*), muscle hypertrophy and cytokine production (e.g. *IL13RA1, IL11RA, CCR7, IL1RAP, IL6ST*).

Latent factor 2: not significantly enriched for any biological processes.

Latent factor 3: regulation of phosphatidylinositol 3-kinase (*ERBB2, KIT, F2R, FLT3, PDGFRB*), regulation of interleukin 4 and 6 (*GFI1, IL6ST, TCF7, LEF1, XCL1*), cell activation (e.g. *TTN, SH2D1B, NCR1, PTPN22, PRKCA*), and T cell activation (e.g. *LEF1, IL6ST, TBX21, VSIG4*).

Latent factor 4: circadian rhythm (most significant enrichment and includes *TIMELESS, ID2, PROK2, GHRL, PER1, PER2, PER3*), response to estrogen (*CITED4, GHRL, SOCS1, SOCS3, BGLAP, LDLR, IL1B, DUSP1*), response to growth hormone (*IRS2, GHRL, SOCS1, SOCS3*) and response to corticosteroid stimulus (*ANXA3, IL1RN, IL1B, BGLAP, DUSP1, SOCS3*). In addition to being enriched for genes involved in circadian rhythmicity, latent factor 4 also contains the temperature-sensitive RNA binding proteins *CIRBP* and *RBM3*, which are known to regulate the translation of circadian transcripts (*Liu et al., 2013*).

Latent factor 5: regulation of plasma lipoprotein clearance (*HNRNPK, CD36, LDLR, APOM*), regulation of interleukin-12 production (*TLR2, TNFSF4, CD36, TIGIT*), and regulation of immune effector process (*TLR2, CADM1, SHSD1B, TNFSF4, CD36, CLCF1*).

Functional enrichment analysis of the 80 genes targeted by the 100 mRNA abundance features identified significantly enriched biological processes associated with leukocyte activation (*HSPH1, SH2D1B, IL6ST, CCND3, CD1C, NCR1, TNFSF4, FLT3, IL1B, TXK*) and inflammatory response (*TLR2, TNFSF4, GHRL, IL1B*).

The one-sample PLSR-based model outperformed all other models discussed so far, for most performance metrics (*Table 1*, *Figure 3*). The average error is comparable to other models but the distribution of error is narrower and the errors did not vary significantly across circadian bins. Fifty-four percent of samples fell within ≤2 hr of the true circadian phase and the $R^2$ of 0.74 also compares well to other models. Finally, the prediction error of the one-sample PLSR model did not differ significantly across the four conditions. Nevertheless, we aimed to produce a better performing model.

## Using more than one sample

One expects the accuracy of a prediction model to increase as the number of input samples increases. We thus applied the one-sample models to pairs and triplets of consecutive samples. Two-samples performed better than one-sample, and three-samples performed better than two (*Table 1*, *Figure 3j,k and l*, *Figure 3—figure supplement 5*). The PLSR-based model gave the best performance with 71% of samples having a prediction error of ≤2 hr and a $R^2$ of 0.88 using three consecutive samples. However, when taking three consecutive samples as input, the model performed significantly worse for samples derived from the condition 'sleeping out of phase with melatonin' (Tukey HSD post-hoc test, p-value<0.05).

## A differential mRNA abundance two-sample circadian phase PLSR-based prediction model

An alternative approach is to develop a model which takes multiple samples as input and utilizes the difference in mRNA abundance between input samples to predict the phases instead of averaging

independent predictions. Thus, it is the difference in the 'state' of the transcriptome, without the significant 'trait-like' characteristic of the transcriptome (*De Boever et al., 2014*) that predicts the phase of an individual. Here, we developed a two-sample based model that takes two-samples taken 12 hr apart in the same participant during the same condition as input. We did this for both the molecular timetable and PLSR approaches (see Materials and methods).

Our trained two-sample differential PLSR model comprises 5 latent factors and 100 features (see *Supplementary file 1D* for the list of features and *Supplementary file 1B* for the list of corresponding genes). The pseudo time-course of factor scores of the five latent factors (*Figure 3—figure supplement 4d*) shows that all are rhythmic ($R^2$ of fit to a cosine wave range of 0.71–0.8), with similar amplitude (range of 0.021–0.022) and maxima phases spanning 360 degrees (250°, 310°, 330°, 320°, 44°, respectively). Functional enrichment analyses of the top 200 features that carry the most weight in each latent factor identified significantly enriched biological processes (see *Supplementary file 2*) including:

Latent factor 1: no significant enrichment.

Latent factor 2: regulation of metabolic process (e.g. *HSPH1*, *ANXA3*, *PPARGC1B*, *LDLR*, *FOSL2*, *CTNNBIP1*, *BCL6*, *PIK3IP1*), regulation of acute inflammatory response (*PTGS2*, *IL1B*, *IL6ST*, *OSM*), response to glucocorticoid stimulus (*ANXA3*, *PTGS2*, *IL1RN*, *IL1B*, *PPARGC1B*, *JUNB*, *SOCS3*) and circadian rhythm (*NR1D1*, *ARNTL*, *PER1*, *PER2*, *PER3*).

Latent factor 3: negative regulation of cellular and metabolic processes (e.g. *DYNLL1*, *ID2*, *NEDD4L*, *DHCR24*, *DUSP1*, *NR2F1*, *AVIL*, *TLR2*), lymphocyte activation (*IRS2*, *CXCR5*, *HSPH1*, *KIF13B*, *CCND3*, *ID2*, *BCL3*, *NDFIP1*, *FLT3*, *IL1B*, *TXK*, *CD83*), rhythmic process (*GHRL*, *TEF*, *ID2*, *TIMELESS*, *ADAMTS1*, *PROK2*, *PER1*, *PER3*) and apoptotic signaling in response to DNA damage (*HIC1*, *FNIP2*, *BCL3*, *DDIT4*).

Latent factor 4: positive regulation of cellular processes (e.g. *CLEC4E*, *INSL3*, *SLC1A3*, *ARG1*, *RBM14*, *RBM3*, *TRAF4*), developmental process (e.g. *CLIC5*, *SSH2*, *SPIB*, *RHBDD1*, *KLF9*, *FKBP4*, *KCNE3*, *SPATA6*), response to lipid (*FADS1*, *TLR2*, *SLC30A4*, *CITED4*, *GHRL*, *BGLAP*, *NFKBIA*, *IRAK3*, *LDLR*, *GATA3*, *IL1RN*, *REST*, *ARG1*, *IL1B*, *DUSP1*) and circadian rhythm (*TIMELESS*, *GHRL*, *PTEN*, *PER1*, *PER2*, *PER3*).

Latent factor 5: regulation of response to stimulus (e.g. *CDKN1C*, *INPP5B*, *OPA1*, *MLC1*, *DVL2*, *APCDD1*, *VSIG4*, *CACNG4*), regulation of immune system process (*KLRD1*, *KIR2DL2*, *SH2D1B*, *A2M*, *CMKLR1*, *NCR1*, *PDE5A*, *KIR2DS2*, *ADA*, *CCL4*, *ADORA2A*, *TBX21*, *VSIG4*, *KLRG1*, *RELA*, *NR1D1*, *APOBEC3G*, *FCN2*), rhythmic process (*ADAMTS1*, *FANCG*, *NR1D1*, *NPAS2*, *BHLHE41*, *PER1*, *ADA*) and calcium-mediated signaling (*NMUR1*, *CHPS*, *CCL4*, *ADA*, *CMKLR1*).

For the two-sample differential PLSR, four of the five latent factors contained genes enriched for circadian rhythm or rhythmic process. Each factor contained at least three clock genes, with factor two containing six genes (*ARNTL*, *NR1D1*, *NR1D2*, *PER1*, *PER2*, *PER3*) together with *CIRBP* and *RBM3*. Aspects of immune function are also common to the factors, but factor five in particular showed enrichment for immune-cell-specific genes (e.g. immune killer cell genes *KIR2DL2*, *KIR2-DL5A*, *KIR2DS2*, *KIR2DS4*, *KLRD1*, *KLRF1*, *KLRG1*), in addition to the interesting inclusion of adenosine deaminase (*ADA*) and the adenosine A2a receptor (*ADORA2A*).

Forty-seven (62%) of the 76 unique genes targeted by the features identified using the two-sample approach also appear in the set of 80 unique genes identified by our one-sample approach (*Supplementary file 1B*). Functional enrichment analysis of the 76 genes identified significantly enriched biological processes including; immune response, circadian and rhythmic processes and regulation of cellular processes. Unlike the one-sample method, the 76-gene list comprising the two-sample differential PLSR model contained all three *PER* genes (together with *NR1D2*), and several genes with immune-cell-specific functions (e.g. *KLRF1*, *KLRD1*, *CD83*, *IL1RN*) .

Both two-sample differential models outperform their preceding one-sample models and multiple consecutive-sample applications (*Figure 4*, *Figure 3—figure supplement 5*, *Table 1*). The PLSR-based model performs well, now able to predict within 20 min on average, with an SD of 1 hr 41 min, without any significant difference in error distribution across circadian phase bins. Significantly, the two-sample differential model is able to correctly predict 82% of samples within 2 hr, with an $R^2$ of 0.9. Similar to the one-sample model, we find that the two-sample differential model is able to predict baseline conditions with higher accuracy than the molecular timetable equivalent. However, unlike the one-sample PLSR model, we find a significant difference in error distribution between conditions, where 'Sleep out of phase with melatonin' has a significantly larger mean error than the

other three conditions (Tukey-HSD post-hoc test p-value<0.05). Nevertheless, the difference in mean error between conditions for our two-sample differential model is less than that observed for the one-sample PLSR model when applied to two or three consecutive samples.

## Discussion

The human blood transcriptome is in part under circadian control and potentially contains information about the phase of circadian rhythmicity in many organs. The analyses presented here show that a few transcriptome samples contain enough information to predict the phase of the melatonin rhythm. This remains true even under conditions in which the sleep-wake cycle, which also influences the blood transcriptome, is desynchronized from the melatonin rhythm. Melatonin phase and shifts in melatonin phase correlate well with other variables driven by the SCN, for example, cortisol (*Dijk et al., 2012*). In clinical practice, it is therefore assumed that the melatonin phase reflects the status of the SCN with minimal error, although the actual error in SCN phase measurement by melatonin is not known. The methods developed here may aid the routine assessment of circadian melatonin phase in clinical practice and benefit the diagnosis and treatment of sleep and circadian disorders (*Mullington et al., 2016*).

### PLSR compared to other approaches

We consider our PLSR-based approach the first method for developing a melatonin phase predictor without *a priori* selection of features or feature characteristics: no *a priori* information is used to supervise model development. We do not select for 24 hr rhythmically expressed genes that are either known in the literature (*Lech et al., 2016*) or specifically identified within the data (*Ueda et al., 2004*; *Hughey et al., 2016*). The validation analyses show that our model comprising features selected *a posteriori* outperforms '*a priori*' approaches. Directly comparing our '*a posteriori*' method with '*a priori*' methods (*Ueda et al., 2004*; *Hughey et al., 2016*), by deploying and testing each using the same training and validation data, shows that our method is valid and capable of generating superior circadian phase prediction models. It should be noted that many of the genes identified by PLSR are not classical clock genes or clock-controlled genes (see *Supplementary file 1B*). Nevertheless, 68% of the genes forming the one-sample, and 63% genes forming the two-sample differential PLSR-based model, we have previously identified as being rhythmically expressed in at least one of the four conditions we have analyzed (*Archer et al., 2014*; *Möller-Levet et al., 2013*; *Laing et al., 2015*). Only 18% and 12% of the one and two sample(s)' genes (respectively) were found to be rhythmically expressed in the condition 'sleep out of phase with melatonin'. In our previous analyses, we used a prevalence-based classification which is a fundamentally different method to the rhythmic detection strategies of the molecular timetable and Zeitzeiger methods (*Ueda et al., 2004*; *Hughey et al., 2016*). However, comparing across prediction models it is clear that >30% (34% and 31% for the one-sample and two-sample differential models, respectively) of the gene lists forming the PLSR-based models are unique to the approach (compared to the molecular timetable and Zeitzeiger approaches applied to the same training data set) (*Figure 5*).

### Molecular process associated with the current feature sets

We used OmicCircos (*Hu et al., 2014*) to visualize and compare the overlap of gene features that were identified by each of the four models trained on the same data (*Figure 5*). Six genes were identified in all four models: *PER1* (circadian clock gene, induced by glucocorticoid [*Cuesta et al., 2015*]), *SMAP2* (involved in vesicle trafficking in the trans-golgi network [*Funaki et al., 2013*]), *DDIT4* (part of mTOR signaling pathway [*Polman et al., 2012*]), *NCR1* (natural killer cell receptor [*Lai and Mager, 2012*]), *ZBTB16* (transcription factor that regulates glucocorticoid-induced gene expression [*Wasim et al., 2010*]) and *FKBP5* (chaperone that plays a key role in the regulation of glucocorticoid receptor signaling [*Binder, 2009*]). Apart from *SMAP2* and *NCR1*, all these genes are involved in glucocorticoid signaling or are regulated by glucocorticoid. Immune function and glucocorticoid signaling are themes that link all four models. More than one third of all the genes in the four models are associated with glucocorticoid signaling (molecular timetable 44%, Zeitzeiger 36%, one-sample PLSR 36%, two-sample differential PLSR 38%) (*Nehmé et al., 2009*).

The gene lists of the molecular timetable and Zeitzeiger models show little overlap with those of the PLSR models, which have a high level of similarity (62%). Fifteen of the top-25 genes (by

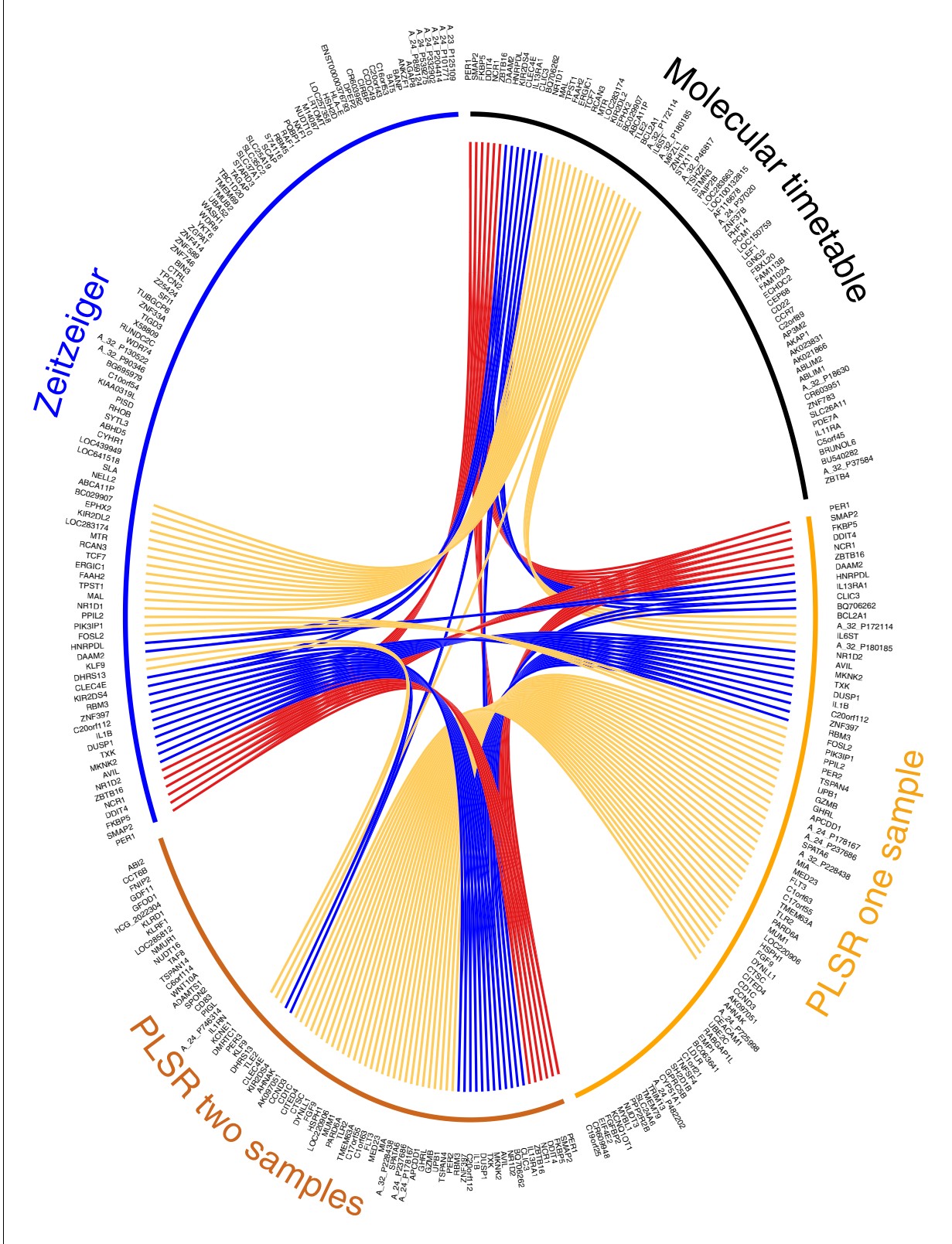

**Figure 5.** Circos plot for visual comparison of lists of biomarkers forming the generated circadian phase prediction models. Circular tracks from outside in; (1) Name of model being compared (Molecular timetable, Zeitzeiger, PLSR one-sample, PLSR two-sample differential), where all models are constructed from the same training set. The color of the model name corresponds to the color of the bar in track 3. (2) Unique HGNC Gene symbols or probe identifiers for features from which the specified model (adjacent track 1 and 3) are created; (3) Color bar indicating the model (Black= Molecular

*Figure 5 continued on next page*

*Figure 5 continued*

timetable, Orange = PLSR one-sample, Dark orange = PLSR two-sample differential, Blue = Zeitzeiger). Inner arcs represent the overlap between models. Genes present in two models are linked by yellow arcs; genes present in three models are linked by blue arcs; and genes present in all models are linked by red arcs. Figure generated using the R package OmicCircos (*Hu et al., 2014*).

The following figure supplement is available for figure 5:

**Figure supplement 1.** A glucocorticoid-driven network links many of the top-ranked PLSR genes.

weighting) in the PLSR gene lists are common to both models (*PER1, DDIT4, FKBP5, ZNF397, TSPAN4, C20orf112, MIA, AVIL, ZBTB16, SPATA6, A_24_P178167, BQ706262, SMAP2, FLT3, HSPH1*). Several of these genes have related functions: MIA and TSPAN4 bind integrin and regulate cell motility (*Bosserhoff, 2005*; *Kashef et al., 2013*), while AVIL binds actin and regulates cell outgrowth (*Tümer et al., 2002*). Twelve of the genes in the top-25 lists are associated with glucocorticoid signaling and form part of a linked glucocorticoid signaling network (*Figure 5—figure supplement 1*). The top four features in the PLSR two-sample differential model gene list are present in all four models (*DDIT4, PER1, FKBP5* and *ZBTB16*).

As discussed, the PSLR approach does not pre-select features with respect to rhythmicity, and indeed cannot be used to detect rhythmicity. Only the feature lists identified by the two sample differential PSLR model appear enriched for rhythmic, circadian processes. This may be because in the differential two-sample approach individual differences in average levels of expression (i.e. trait like characteristics of the transcriptome [*De Boever et al., 2014*]) are removed. This apparently emphasizes features with a 24 hr variation around a baseline.

Both the cortisol and the melatonin rhythm are driven by the SCN. The prominence of glucocorticoid signaling in the lists may reflect a more extensive influence of cortisol on the peripheral transcriptome. Overall the features of the PLSR models seem to focus more on genes with functions specifically related to blood such as glucocorticoid signaling and immune cell-specific roles. Thus, the features within the blood transcriptome that are most predictive of the melatonin phase are closely related to blood-cell-specific functions. We have previously shown that the influence of the sleep-wake cycle and circadian rhythmicity varies across the transcriptome transcripts in blood (*Archer et al., 2014*). Whether and how these blood-cell-specific functions remain in phase with the SCN, even when the sleep-wake cycle is shifted, remains unclear. It may be that those that predict SCN phase are regulated by local circadian clocks, which in turn are synchronized by systemic cues from the central circadian clock, such as cortisol. Alternatively, these SCN phase predictors in blood may be directly driven by systemic cues such as cortisol or melatonin. As far as melatonin is concerned, only very few melatonin-related genes appeared in any of the models. The circadian expression of pineal HNRNP proteins regulates the expression of the rate-limiting melatonin synthesis enzyme AANAT (*Kim et al., 2005*). In this study, *HNRPDL* is a feature of each model except the PLSR two-sample differential model and was previously identified as remaining rhythmic in all four of the sleep conditions that have contributed data to this study (*Laing et al., 2015*). Thus, while glucocorticoid signaling features strongly in the models developed here, it is also possible that melatonin signaling contributes to phase entrainment in whole blood, possibly also locally within blood cells.

## Limitations of the current study

The enhanced power of our PLSR-based models compared to others cannot be due to the unique combination of experimental conditions that we have included in our training and validation sets because all models were tested on the same data sets. Our training and validation data are limited; all samples were collected in darkness and dim light and from young healthy individuals. Comparing the performance of the various models across conditions emphasizes the importance of inclusion of conditions in which synchrony between various organs is challenged. Several models that performed well under normal conditions performed less well under desynchronized conditions. Indeed, we expect the conditional dependency of performance for the two-sample differential models to be due to the significant temporal disruption evoked by sleeping out of phase with melatonin, where the difference in mRNA abundance between two samples taken 12 hr apart may not be so well preserved as in baseline and/or sleep deprivation conditions. Thus, inclusion of similar conditions will be

important for the validation for any blood-based biomarker of organ-specific rhythmicity and the generation of clinically relevant applications in 'real-world' scenarios. Patients may be suffering from sleep deprivation or sleep and food-intake desynchronized from the circadian system, conditions in which rhythmic molecular profiles are disrupted. A 'final' model to be applied in the clinic should also be trained on real conditions in which light exposure and food availability are not controlled to ensure its validity to other sections of the population. This could include not only circadian rhythm sleep-wake disorder patients but also other populations in which melatonin phase and or cortisol phase are disrupted, such as in Smith-Magenis syndrome (*De Leersnyder, 2013*) or in Addison's disease (*Oster et al., 2016*).

## Outlook

Several non-omics-based methods to assess circadian phase are currently in use. In the Dim-Light-Melatonin Onset (DLMO) Protocol, many samples (6–18) are collected during an approximately 6–12 hr period timed to be close to the expected onset of melatonin secretion (*Burgess et al., 2016*). Thus, unlike the method presented here, the DLMO protocol requires prior knowledge of the approximate circadian phase of the pacemaker to identify the phase of the SCN (with unknown error). In fact, one immediate application of the current model is to use it to identify the time window during which a DLMO assessment should be scheduled. Alternative assessments of SCN circadian phase, when there is no prior knowledge about the likely circadian phase, requires measurements covering at least one circadian cycle. In field situations, this can be accomplished by the scheduled collection of urine for determination of 6-sulfatoxy-melatonin, where the 95% CIs of such assessment are thought to be ±1 hr 30 min (*Lushington et al., 1996*). Yet, even when urine samples are collected very frequently (e.g. every 4 hr), the correlation between the urine-based phase marker 6-sulfatoxy-melatonin and the gold-standard plasma-based phase marker melatonin is only 0.75 (*Gunn et al., 2016*). Similarly, accurate assessment of circadian phase from core body temperature in the presence of a sleep-wake cycle is problematic because sleep and wakefulness have a direct effect on core body temperature, which depends on circadian phase. Therefore, extraction of phase of the circadian pacemaker from core body temperature requires either recordings over several days or weeks, or 24–40 hr T recordings in specialized laboratory conditions.

From a practical perspective, a circadian phase predictor that requires only one sample is ideal, minimizing the total time in the clinic and overall cost. However, the marked increase in performance offered by taking multiple samples, either two or three consecutive samples 3 or 4 hr apart, or two-samples 12 hr apart for differential analysis, should also be considered. Certainly, the multiple-sample models/applications we tested here rely on far fewer samples than 6-sulfatoxy-melatonin and/or core body temperature, even plasma melatonin assessments, and can be completed during one 'out-patient' clinic visit. Our PLSR-based models and applications can predict more than 80% of circadian phases within 2 hr of the observed phase when using the differential analysis of two-samples as input.

The current models are based on whole-blood transcriptomic profiles comprising mRNA abundance features measured by microarrays. However, the methodology we present can be readily applied to map any high-dimensional molecular abundance 'feature' profile, such as those generated by microarray, RNA sequencing and metabolomics technologies, to any specified measure of circadian phase. Although we used melatonin/SCN phase as an example with obvious applications in circadian rhythm sleep-wake disorders, one can imagine in the era of precision medicine the development of specific PLSR-based circadian phase prediction models for the liver and/or other specific tissues and organs. All these organs may be targeted by chronotherapy (*Zhang et al., 2014*). All that is required is the identification of specific phase markers for the tissue/organ of clinical interest.

Transcriptome-based measures appear to suffer from less noise than other '-omes' (*Su et al., 2011*; *Tachibana, 2014*). The 100 transcriptomic features we have identified as predictors of SCN circadian phase using whole-blood can be simultaneously measured by a cost-effective low-resolution molecular technique. Therefore, the PLSR-based models we present here offer a viable solution for a clinical platform (*Mullington et al., 2016*); even more so when redundant features that target the same gene/transcript, are removed.

## Materials and methods

### Source of data

Data were collected during two large-scale circadian experiments to investigate the effects of mistimed sleep and total and repeated partial sleep deprivation on the temporal organization of the human blood transcriptome (*Archer et al., 2014*; *Möller-Levet et al., 2013*). The protocols received a favorable opinion from the University of Surrey's Ethics committee and all participants provided written informed consent. In each experiment, we obtained from the same participant, blood samples (every 3 or 4 hr) covering at least one circadian cycle to measure mRNA abundance and hourly blood samples for measuring melatonin rhythms. This allowed assignment of a circadian melatonin phase to each mRNA sample. Melatonin concentrations were determined with a radioimmunoassay (Stockgrand Ltd, Guildford, UK [*Fraser et al., 1983*]). Circadian phase was defined as the time when melatonin levels reached 25% of the melatonin amplitude and is referred to as the Dim Light Melatonin Onset (DLMO). DLMO = (((melatonin amplitude - baseline melatonin level)*25)/100) + baseline melatonin level (*Hasan et al., 2012*; *Lo et al., 2012*).

We collected 678 samples for which we have both the melatonin phase and mRNA abundance profile from 53 participants. Fifty-nine percent (397) of these samples are from male participants (see *Supplementary file 3* for demographics, including ethnicity). The participant population was stratified according to the *PER3* rs57875989 genotype (38% $PER3^{4/4}$; 35% $PER3^{5/5}$; 27% $PER3^{4/5}$). The complete time-series of samples from homozygote participants were hybridized to microarrays (i.e. 14–20 samples per participant; 7–10 samples per sleep condition). Select RNA samples (one to three samples per sleep condition) from the complete time-series of samples for heterozygote individuals were hybridized to microarrays. All transcriptomic data are accessible from the Gene Expression Omnibus (*Barrett et al., 2013*) (RRID:SCR_007303, via the Accession numbers: GSE48113 [*PER3* homozygote, $PER3^{4/4}$ and $PER3^{5/5}$] and GSE82113 [*PER3* heterozygote, $PER3^{4/5}$] data for mistimed sleep, and GSE39445 (homozygote) and GSE82114 (heterozygote) data for sleep deprivation and training) and the data sources at http://sleep-sysbio.fhms.surrey.ac.uk/PLSR_16/.

### Creation of training and validation data sets

Samples were split into a training set to develop the models, and a validation set to test the predictive performance of the models. Factors that were balanced between the training and validations sets were: the experimental protocol (mistimed sleep or sleep deprivation), *PER3* genotype, sex and circadian phase (as indicated in *Supplementary file 3*). All samples pertaining to a participant were assigned to the same data set. Data of 26 participants, comprising 329 samples were randomly assigned to the training set and the data of 27 participants, comprising 349 samples to the validation set. To facilitate the development and benchmarking of future models, the data matrices for the training and validation data sets are provided at: http://sleep-sysbio.fhms.surrey.ac.uk/PLSR_16/.

For models developed and tested for a 'one sample' application, including the prediction of consecutive samples requiring the application of a one-sample model multiple times, all 329 and 349 samples of the training and validation sets were used. For 'two-sample differential' models the mRNA abundance values for a pair of samples from the same participant during the same condition, separated by 12 hr, were reduced to a single observation. The single observation represents the difference in the mRNA abundance, that is, a vector obtained as a result of computing Sample one minus Sample 2 (taken 12 hr later than Sample 1). In total, the 'two sample differential' training and validation data sets comprised 163 and 180 mRNA abundance difference 'samples', respectively. The data matrices for the training and validation data sets are provided at: http://sleep-sysbio.fhms.surrey.ac.uk/PLSR_16/. Note that the two-sample differential molecular timetable model is based on the one-sample molecular timetable. For this two-sample model, the difference in the timetable templates with a 12 hr time lapse is used to predict the difference in mRNA abundance between two samples taken 12 hr apart.

### Data processing

For the training set the $Log_2$ mRNA abundance values were processed by applying quantile-normalization using the R Bioconductor package limma (*Gentleman et al., 2004*; *Smyth, 2005*) (RRID:SCR_010943) followed by batch correction using the ComBat function of the R package sva (*Leek et al.,*

*2012*; *Johnson et al., 2007*) (RRID:SCR_012836). For the validation set, the $Log_2$ mRNA abundance values were quantile-normalized using the same reference array of empirical quantiles used to normalize the samples included in the training set. No batch correction was applied to the validation data set to reflect the type of data that would be provided in a clinical setting (i.e. one or two samples processed in different batches to the original training data used to build the model). Non-control technically replicated probes (mRNA abundance features), along with their corresponding Agilent feature flags (Agilent Feature Extraction Software [v10.7], Reference Guide, publication number G4460-90036) were averaged. Individual samples with more than 30% of features flagged were excluded and features (probes) flagged in more than 10% of the samples were filtered out.

## Directional mean

Assuming angles lay on a unit circle (radius = 1), an angle $\theta$ can be converted to Cartesian coordinates $(x, y)$ using the trigonometric functions of $y = sin\theta$ and $x = cos\theta$. The Cartesian coordinates $(x, y)$ can be converted back to an angle $\theta$ using the arctan function $\theta = tan^{-1}(y/x)$ for $(x, y)$ in quadrant I of the coordinate plane, $\theta = tan^{-1}(y/x) + \pi$ for $(x, y)$ in quadrant II or III, and $\theta = tan^{-1}(y/x) + 2\pi$ for $(x, y)$ in quadrant IV of the coordinate plane. In this way, the mean value $(\bar{\theta})$ of a set of $n$ angles $(\theta)$ can be calculated as

$$\bar{\theta} = \tan^{-1}(\bar{y}/\bar{x})\ 180/\pi + q \tag{1}$$

where $\bar{y} = \sum_{i=1}^{n} \sin(\theta_i\ \pi/180)/n$, $\bar{x} = \sum_{i=1}^{n} \cos(\theta_i\ \pi/180)/n$, and $q = 0$ if $(\bar{x}, \bar{y})$ is in quadrant I of the coordinate plane, $q = 180$ if $(\bar{x}, \bar{y})$ is in quadrant II or III and $q = 360$ if $(\bar{x}, \bar{y})$ is in quadrant IV.

## Difference between two angles

The difference $\delta$ between angles $\theta$ and $\beta$ can be calculated using

$$
\begin{aligned}
\delta_{\theta\beta} &= \theta - \beta & \text{if} \quad -180 < (\theta - \beta) < 180, \\
\delta_{\theta\beta} &= -(360 - (\theta - \beta)) & \text{if} \quad 180 < (\theta - \beta) \\
\delta_{\theta\beta} &= (\theta - \beta) + 360 & \text{if} \quad (\theta - \beta) < -180.
\end{aligned}
$$

## Measuring model performance

For all operations, angles $\theta$ were converted to Cartesian coordinates (*cosine* $\theta$, *sine* $\theta$) to perform arithmetic operations.

To assess the overall performance and clinical relevance of our models, we used several metrics:

1. The frequency distribution of error between the predicted circadian phase $\hat{y}$ and the experimentally observed circadian phase $y$, when binned in 15° (60 min) circadian phase bins. The distribution is summarized by the mean and standard deviation.
2. The fit of predicted phases to the observed phase via $R^2$. If $y_i$ is the observed circadian phase and $\hat{y}_i$ the predicted circadian phase of the $i^{th}$ sample, $R^2$ is defined as:

$$R^2 = 1 - \frac{RSS}{TSS} = 1 - \frac{\sum_{i=1}^{n}(y_i - \hat{y}_i)^2}{\sum_{i=1}^{n}(y_i - \bar{y})^2} \tag{2}$$

where $\bar{y}$ is the directional mean of the observed circadian phases, RSS: regression sum of squares; TSS: total sum of squares. Thus, $R^2$ compares the variance in prediction error to the variance in the observed circadian phases, giving a fraction of unexplained variance. When subtracted from 1, the $R^2$ value represents the variance that can be explained by the model. Where RSS (variance in prediction error) is greater than TSS, the resultant $R^2$ will have a negative value, corresponding to a model that performs worse than fitting a horizontal line.

3. The error distribution across the sampling period of 24 hr (360 degrees), where a better performing model is one that exhibits no (or least) dependency on the time at which the sample is taken. Differences in error across time was assessed by ANOVA (Analysis of Variance) analysis for the factor of 'sampling time bin' using the lm and anova functions in R (RRID:SCR_001905); a p-value<0.05 suggests that the prediction accuracy of the model is dependent on the circadian phase at which the sample is taken.
4. The cumulative frequency distribution of absolute error, to identify the proportion of samples that can be predicted within a given error range, where the highest proportion combined with smallest error range denotes the better prediction model.

To identify parameters and/or thresholds with which to build a model and/or assess the overall performance of the final model, we used the performance metric $R^2$. The exception to this is the Zeitzeiger model which relies on the Mean Average Error to identify parameters (see below).

## Assessing circadian phase prediction performance across conditions

Differences in circadian phase prediction errors across the four conditions were assessed using the 'aov' function of R followed by Tukey Honest Significant Difference (HSD) post-hoc test (R function TukeyHSD) (RRID:SCR_001905).

## Cross validation

To avoid over-fitting during model development, we performed leave-one-participant-out cross-validation. For $p$ participants in the original one or two-sample training set, all samples associated with $p$-1 participant were used for initial training of the model, and the phase of all samples associated with the excluded participant were used to test the model. Once all $p$-1 sets were used to create and test the model the predicted phases were used to determine the predictive power of the cross-validated model, as described above.

## Model development and implementation

All code used to build and test the models, calculate the given statistics and generate the provided figures, along with a description on their usage can be found at: http://sleep-sysbio.fhms.surrey.ac.uk/PLSR_16/.

## Molecular timetable

We implemented the molecular timetable method described in *Ueda et al. (2004)*. Instead of using a summary metric (e.g. mean or median) or model to represent time-series across participants, we used each individual sample within the four experimental conditions in the training set to generate a single, densely-sampled time-series per mRNA abundance feature per experimental condition. We then created a fifth, 'all conditions' time-series, by using each individual sample in the training set irrespective of experimental condition. Subsequently, we selected the mRNA abundance features with correlation values to the cosine templates above the cutoff correlation value of 0.3 for all five time-series. The value of 0.3 was identified as the optimal threshold as it produced the highest $R^2$ value following leave-one-participant-out cross-validation using different correlation thresholds. *Figure 2—figure supplement 1* depicts the change in model performance ($R^2$ values) at different correlation thresholds. It was not necessary to use the coefficient of variation to further filter the time-indicating features given that only a small fraction of all features have a detectable circadian oscillation under all experimental conditions included in our analyses. Where multiple mRNA features targeting the same gene met our correlation threshold, the best performing (most correlated to a cosine) mRNA feature was selected to represent a gene. Subsequently, the method identified 73 mRNA abundance features, which target 73 unique genes, as time-indicating features.

The timetable model is essentially a look-up table. To predict the phase of a single sample, the set of mean-centred mRNA abundance values for the 73 features comprising the model are extracted and compared to the expected set of values at different phases within the molecular timetable model (look-up table). The phase producing the highest correlation is the sample's predicted melatonin phase. In a similar approach, the resulting molecular timetable was further used to construct a 'two-sample' molecular timetable, where the difference in mRNA abundance of two samples taken 12 hr apart is compared to the difference in the timetable templates with the same time lapse.

## Zeitzeiger

We used the Zeitzeiger method of (*Hughey et al., 2016*) implemented in the zeitzeiger R package, downloaded from https://github.com/jakejh/zeitzeiger (zeitzeiger_0.0.0.9000, RRID:SCR_014791). To build a Zeitzeiger predictor, given a training data set, there are two parameters that need to be optimized: 'sumabsv', which controls for the overfitting of the model by controlling the number of features (probes measuring mRNA abundance) to be used to form principal components, and 'nSPC', which controls the number of principal components that describe how the features change

over time. To identify the best parameters for training the predictor, we ran leave-one-participant-out cross-validation of the training set. The cross-validation was run over seven values of sumabsv (1, 1.5, 2, 2.5, 3, 3.5 and 4) and five values of nSPC (1, 2, 3, 4 and 5). To compare the performance of the predictors generated from each set of parameter values, we calculated the Mean Average Error (MAE). As seen in *Figure 3—figure supplement 2*, the smallest MAE is produced by the combination of sumabsv = 4 and nSpc = 3. These optimal values were then used to produce a Zeitzeiger based model to predict the circadian phase of samples in the validation set, with which we could assess the overall performance of the model and directly compare with other models. The 107 mRNA abundance features (probes) that can predict circadian phase based on the Zeitzeiger approach were identified as those carrying weight in at least one of the three principal components used to construct the model.

## Partial least squares regression

In this work, $X \in \mathbb{R}^{mxn}$ is a matrix of $m$ samples by $n$ mRNA abundance features and the vectors $(Y_x, Y_y) \in \mathbb{R}^m$ are the corresponding observed melatonin phase of each sample represented in Cartesian coordinates. Partial Least Squares Regression (PLSR) is a linear regression model that regresses $(Y_x, Y_y)$ on to $X$ by projecting both $(Y_x, Y_y)$ and $X$ into a reduced-dimensional space (so called latent factor space) that maximizes their covariance before performing a least squares regression. This projection to a latent space enables PLSR to handle the observed co-linearity between mRNA abundance features derived from the same gene, that is, multiple probes targeting the same transcript and/or co-expressed 'modules'. To build a PLSR model, given a training data set, there are two parameters that need to be optimized: (1) the number of latent factors, that is dimensions, $T$ $(1<T<N)$ and (2) the number of mRNA abundance features $n$ $(1<n<N)$. If the values of these parameters are set too high (e.g. maximum value), there is a danger of over-fitting the model to the training set, and if the parameters are set too low, the model would not be sensitive nor specific enough to provide high accuracy. Similar to the training in the molecular timetable and Zeitzeiger methods, above, we used leave-one-participant-out cross-validation of the training set to identify the best combination of model parameters, in this case the number of latent factors and number of mRNA abundance features. For our one-sample PLSR approach, we compared the cross-validated performance of 136 combinations of 5 to 40 latent factors (in increments of 5) and 100 to 5000 features with which to build a model (see *Figure 3—figure supplement 4a*). For our two-sample PLSR approach, we compared the cross-validated performance of 100 combinations of four to eight latent factors and 50 to 1000 features (*Figure 3—figure supplement 4c*). To generate each model, the plsregress function of Matlab (8.2.0.701 [R2013b]) was invoked twice. The first call to the plsregress function generates a PLSR-based model given the z-scored mRNA abundance feature values (for all features) of the training set (where samples for a single-participant have been removed), the corresponding melatonin phase values represented in Cartesian coordinates, and the number of specified latent factors as input. The absolute loadings (weights) of this model are then used to identify the top $n$ features to create a subset of the training set comprising the $n$ features. This subset of the training set used in the first call is then used as input to the subsequent second call of the plsregress function given the same (as the preceding call to the function) melatonin phase values represented in Cartesian coordinates, and the number of specified latent factors as input. The model derived from the second call to the plsregress function is then used to predict the melatonin phase of the left out (cross-validation) samples. Note that each $p$-trained model during cross-validation may have a different set of 'top' (most weighted) features as feature order was not fixed in the cross-validation. The performance of a combination of latent factors $T$ and features $n$ is then assessed by R$^2$ between the predicted and true melatonin phase values across all 'leave-one-out' predictions from models comprising latent factors and features. The overall optimal combination of parameters was defined as the combination that produced an R$^2$ >= 70% using the least number of features and latent factors. *Figure 3—figure supplement 4* depicts the changes in performance (R$^2$) of the model as the number of features and latent factors change when using one sample (*Figure 3—figure supplement 4a*) or two samples (*Figure 3—figure supplement 4c*) as input. As can be seen in *Figure 3—figure supplement 4a*, when predicting circadian phase using one transcriptomic sample one of the highest R$^2$ values, at 73%, occurs when using 100 mRNA features and five latent factors. The maximum R$^2$ value of 76% is obtained by increasing the features to 2000 and 10 latent

factors. However, given that this number of features could induce overfitting and affect clinical applications (e.g. too many features to measure in a single platform) without a significant increase in $R^2$ we chose the combination of 100 mRNA features and five latent factors as an acceptable combination of parameters with which to build the final model. Using the same approach, for two samples (see *Figure 3—figure supplement 4c*), we identified 100 features and five latent factors ($R^2$ value of 93%) as acceptable parameter values with which to train the final predictive model.

### Future deployment of our PLSR-based models

Provided that a future user has blood transcriptome mRNA abundance values from the same micro-array platform (GEO [*Barrett et al., 2013*; RRID:SCR_007303] accession number GPL15331) and/or features, our models can be used to predict the circadian phase of the participant from which the sample(s) were taken. The raw $Log_2$ mRNA abundance values of the test sample(s) will need to be quantile-normalized using the same reference array of empirical quantiles used to normalize the samples included in our training set (see data at http://sleep-sysbio.fhms.surrey.ac.uk/PLSR_16/).

The PLSR method projects both the predicted and the observed variables onto a new space to find a linear regression model. The linear regression model is used to predict circadian phase ($\hat{y}$) from a set of $n$ mRNA abundance values ($x$), where $\hat{y}$ is calculated from the predicted Cartesian coordinates of ($\hat{y}_c, \hat{y}_s$). Thus, the model parameters are the $n$ regression coefficients ($w_c$) and the intercept value ($w_{c0}$) for the cosine of the predicted phase ($\hat{y}_c$), such that $\hat{y}_c = \sum_{i=1}^{n} w_{ci}x_i + w_{c0}$, and the $n$ regression coefficients ($w_s$) and the intercept value ($w_{s0}$) for the sine of the predicted phase ($\hat{y}_s$), such that $\hat{y}_s = \sum_{i=1}^{n} w_{si}x_i + w_{s0}$. The predicted phase is then $\hat{y} = tan^{-1}(\hat{y}_s/\hat{y}_c)\,180/\pi + A$, where $A = 0$ if ($\hat{y}_c, \hat{y}_s$) is in quadrant I of the coordinate plane, $A = 180$ if ($\hat{y}_c, \hat{y}_s$) is in quadrant II or III and $A = 360$ if ($\hat{y}_c, \hat{y}_s$) is in quadrant IV.

A circadian phase prediction can therefore be obtained by computing the sum of the products of the $n$ normalized z-scored mRNA abundance values and their corresponding regression coefficients and adding the intercept values (see data at http://sleep-sysbio.fhms.surrey.ac.uk/PLSR_16/).

### Repeated use of one-sample based models

The mean true circadian phase of two or three consecutive samples was directly compared with the mean predicted circadian phase of two or three consecutive samples, as calculated using a one-sample model, for example, molecular timetable, Zeitzeiger and PLSR.

### Functional enrichment analysis

Functional enrichment analysis was performed using the online tool Webgestalt (*Wang et al., 2013*) (RRID:SCR_006786). Here, a given mRNA abundance feature list was converted to a list of unique HGNC gene symbols and uploaded to the Webgestalt tool. Gene Ontology (GO) enrichment analysis was performed using the reference human genome as the background and selecting for significant (Benjamini and Hochberg adjusted p-value<0.05) GO biological processes and GO molecular functions.

## Acknowledgements

We thank Drs Alpar Lazar, June Lo, Sibah Hasan, Emma Arbon, research nurses, research officers, program managers, and study physicians of the Surrey Clinical Research Centre for data acquisition and clinical support. We thank Malcolm von Schantz, Jonathan Johnston, Colin Smith, Giselda Bucca and John Groeger for helpful discussions and Benita Middleton for melatonin assays.

## Additional information

### Funding

| Funder | Grant reference number | Author |
| --- | --- | --- |
| Air Force Office of Scientific Research | FA9550-08-1-0080 | Emma E Laing<br>Simon N Archer<br>Derk-Jan Dijk |

| Biotechnology and Biological Sciences Research Council | BB/F022883 | Simon N Archer Derk-Jan Dijk |
| Medical Research Council | MR/M023281 | Norman Poh |
| Royal Society | | Derk-Jan Dijk |

The funders had no role in study design, data collection and interpretation, or the decision to submit the work for publication.

## Author contributions

EEL, Conceptualization, Resources, Data curation, Software, Formal analysis, Supervision, Funding acquisition, Investigation, Visualization, Methodology, Writing—original draft, Writing—review and editing; CSM-L, Conceptualization, Resources, Data curation, Software, Formal analysis, Funding acquisition, Investigation, Visualization, Methodology, Writing—original draft, Writing—review and editing; NP, Conceptualization, Resources, Software, Formal analysis, Investigation, Methodology, Writing—original draft, Writing—review and editing; NS, Resources, Data curation, Funding acquisition, Methodology, Writing—original draft, Writing—review and editing; SNA, Funding acquisition, Investigation, Methodology, Formal Analysis, Visualization, Writing—original draft, Writing—review and editing; D-JD, Conceptualization, Formal analysis, Supervision, Funding acquisition, Investigation, Methodology, Writing—original draft, Project administration, Writing—review and editing

## Author ORCIDs

Emma E Laing, http://orcid.org/0000-0002-2095-2442

## Ethics

Human subjects: The protocols used to produce the data used in this work received a favourable opinion from the University of Surrey's Ethics committee and all participants provided written informed consent.

# Additional files

## Supplementary files

• Supplementary file 1. Comparison of phase marker lists. (A) Correlation *r* values and relative rank of correlation value for genes in phase marker lists. Maximum correlation for a gene is based on the maximum correlation between the temporal profile of a feature targeting that gene and a cosine wave. Temporal profiles were constructed independently for each condition and across all conditions. Rank of a gene is based on the distribution of maximum *r* values for a specific condition. Columns in the file; (A) Probe name; (B) Gene Symbol (or probe name if no gene is assigned); (C) Binary values identifying a gene as present (1) or absent (0) in the list of genes forming the molecular timetable model generated here; (D) Binary values identifying a gene as present (1) or absent (0) in the list of genes forming the model of (*Lech et al., 2016*); (E) Binary values identifying a gene as present (1) or absent (0) in the list of genes forming the model of (*Hughey et al., 2016*); (F) Binary values identifying a gene as present (1) or absent (0) in the list of genes forming the Zeitzeiger model generated here; (G) The maximum correlation *r* value of a gene across all four conditions used in this study; H) The maximum correlation *r* value of a gene in the condition 'sleep in phase with melatonin'; (I) The maximum correlation *r* value of a gene in the condition 'sleep out of phase with melatonin'; (J) The maximum correlation *r* value of a gene in the condition 'total sleep deprivation, no prior sleep debt'; (K) The maximum correlation *r* value of a gene in the condition 'total sleep deprivation, prior sleep debt'; (L), (M), (N), (O), and (P) provide the ranking of the correlation *r* value in the corresponding condition(s) of columns (G), (H), (I), (J) and (K) respectively. (B) Comparison of gene lists derived from different phase marker models and/or analyses. Genes identified in at least one of the gene lists discussed in this work (as indicated by the key within the file). A value of 1 indicates presence in the list, a value of 0 indicates absence. (C) Features (probes) and corresponding gene symbols for the one-sample PLSR model. (D) Features (probes) and corresponding gene symbols for the two-sample differential PLSR model.

• Supplementary file 2. Results table for Functional enrichment analysis of feature lists and latent factors for both the one-sample and two-sample differential PLSR-based models. Functional enrichment analysis outputs from using the Webgestalt functional enrichment analysis tool.

• Supplementary file 3. Demographic information for the participants within the training and validation data sets.

## Major datasets

The following datasets were generated:

| Author(s) | Year | Dataset title | Dataset URL | Database, license, and accessibility information |
| --- | --- | --- | --- | --- |
| Moller-Levet C, Archer S, Bucca G, Laing E, Slak A, Kabiljo R, Lo J, Groeger J, Santhi N, von Schantz M, Smith CP, Dijk DJ | 2017 | Mistimed sleep disrupts circadian regulation of the human blood transcriptome - Heterozygotes | http://www.ncbi.nlm.nih.gov/geo/query/acc.cgi?acc=GSE82113 | Publicly available at the NCBI Gene Expression Omnibus (accession no: GSE82113) |
| Moller-Levet C, Archer S, Bucca G, Laing E, Slak A, Kabiljo R, Lo J, Groeger J, Santhi N, von Schantz M, Smith CP, Dijk DJ | 2017 | The human blood transcriptome following sleep extension and sleep restriction: Heterozygote samples | http://www.ncbi.nlm.nih.gov/geo/query/acc.cgi?acc=GSE82114 | Publicly available at the NCBI Gene Expression Omnibus (accession no: GSE82114) |

The following previously published datasets were used:

| Author(s) | Year | Dataset title | Dataset URL | Database, license, and accessibility information |
| --- | --- | --- | --- | --- |
| Moller-Levet C, Archer S, Bucca G, Laing E, Slak A, Kabiljo R, van der Veen D, Lazar A, Groeger J, Santhi N, von Schantz M, Smith CP, Dijk DJ | 2014 | Mistimed sleep disrupts circadian regulation of the human blood transcriptome | http://www.ncbi.nlm.nih.gov/geo/query/acc.cgi?acc=GSE48113 | Publicly available at the NCBI Gene Expression Omnibus (accession no: GSE48113) |
| Moller-Levet C, Archer S, Bucca G, Laing E, Slak A, Kabiljo R, van der Veen D, Lazar A, Groeger J, Santhi N, von Schantz M, Smith CP, Dijk DJ | 2013 | Effect of sleep restriction on the human transcriptome during extended wakefulness | http://www.ncbi.nlm.nih.gov/geo/query/acc.cgi?acc=GSE39445 | Publicly available at the NCBI Gene Expression Omnibus (accession no: GSE39445) |

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
