## [Decision Letter]

Thank you for submitting your article "Blood transcriptome based biomarkers for human circadian brain pacemaker time" for consideration by *eLife*. Your article has been reviewed by three peer reviewers, and the evaluation has been overseen by a Reviewing Editor and a Senior Editor. The following individuals involved in review of your submission have agreed to reveal their identity: Pål O Westermark (Reviewer #1); John B Hogenesch (Reviewer #3).

The reviewers have discussed the reviews with one another and the Reviewing Editor has drafted this decision to help you prepare a revised submission.

Summary:

Laing and co-authors developed an unbiased circadian phase prediction method using partial least squares regression (PLSR) to model the relationship between whole-blood mRNA abundance profiles and their corresponding melatonin phase. Knowing body time is increasingly important for drug development and delivering medicines at optimal times for treatment. In this paper, the authors collected published datasets studying circadian transcription and plasma melatonin levels in human blood samples under different sleep conditions. Using these data, the authors compared published methods to predict body time with transcripts. The authors also developed their own partial least squares regression (PLSR) method to predict body time. Compared with the Molecular timetable and ZeitZeiger methods, PLSR better predicts circadian phase with blood transcripts.

In principle, the topic of the paper is important and the findings are novel and PLSR seems to be an advance; however, there was no statement regarding the availability of the PLSR code or information on what language it was written. Without this, it will be difficult to independently confirm the authors' work, or have the larger research community make use of the method. This reduced the enthusiasm for the paper among the reviewers.

Essential revisions:

1) Because the paper is long and so verbose, the main point of this paper is hard to understand. All of the main texts and figures should be more concise. Many figures are of technical nature and should perhaps be shown in the supplement. In the present manuscript version, the important messages drown in a wealth of material of more technical nature.

2) The SCN phase is not the same as the melatonin phase, the authors describe "predicting the circadian phase of the SCN" throughout the manuscript. As the authors described in Discussion, actual error in SCN phase measurement by melatonin is not known. The authors should use "melatonin phase" instead of "SCN phase". In addition, knowing the melatonin phase is not a panacea for delivering drugs at appropriate times for all diseases (for example, knowing the phase of the liver for a liver-specific disease may be more relevant than knowing the melatonin phase). The authors need to add discussion for this point.

3) The authors should make the code available (e.g. an R package, e.g. ZeitZeiger, or Python or whatever). In addition to making the code available, a more detailed description of PLSR should be included.

4) The authors did a good job of pointing to the primary data. However, they should also document the code/scripts used to run the Molecular timetable method, ZeitZeiger, and PLSR so that others can replicate this analysis.

5) What influence does the size of the training set have on PLSR's accuracy?

6) Can PLSR be used to rule out rhythmicity (ala Molecular timetable and ZeitZeiger)?

7) The Molecular timetable method has also been used with metabolite data. Does PLSR work with metabolite data as well?

8) From Table 1 and Figure 4, PLSR under-estimates melatonin phase (negative error value). Is this correct?

9) The Agostinelli et al. paper is cited. BIO_CLOCK should be added to the algorithms benchmarked here.

10) This model improves upon previous models, but what advantage does this improved model have over previous models? Or, what advantage does this model have over just measuring melatonin? For example, what disease would benefit from having a model that has 20% error compared to 10% error? Of course, having a more accurate model is better, but it's not clear what problems or challenges that this new model will solve. Or what issues could be addressed that were not addressed by previous models?

11) The SD of the prediction error is on the order of 2-3 hrs. Is this likely to be enough for the method to be practically useful in e.g. clinical applications?

12) Authors should explain why they think only PER1 was identified in both molecular timetable methods (subsection “Molecular Timetable and Zeitzeiger methods”, second paragraph).

13) Authors should explain why they think those particular 4 genes were identified as circadian in all four conditions (subsection “Molecular Timetable and Zeitzeiger methods”, second paragraph). Because other hnRNPs can directly regulate AANAT, the rate-limiting step in melatonin biosynthesis, is it possible that hnRNPDL is also directly regulating AANAT (as opposed to, or in addition to, NRF as the authors suggest in their model)? And could this regulation occur via locally produced melatonin from peripheral blood mononuclear leucocytes?

14) In Figure 2, the cutoff correlation value of 0.3 is slightly arbitrary because there is so much variation in model performance with slightly different correlation values. For example, a cut-off correlation value of 0.4 performs almost as well as 0.3. Authors should discuss about this point.

---

## [Author Response]

*In principle, the topic of the paper is important and the findings are novel and PLSR seems to be an advance; however, there was no statement regarding the availability of the PLSR code or information on what language it was written. Without this, it will be difficult to independently confirm the authors' work, or have the larger research community make use of the method. This reduced the enthusiasm for the paper among the reviewers.*

For the implementation of PLSR we have used the ‘plsregress’ function in Matlab [8.2.0.701 (R2013b)]. All code used to build and test the models, calculate the given statistics and generate the provided figures are now provided, along with a description on their usage, on our server at: http://sleep-sysbio.fhms.surrey.ac.uk/PLSR_16/.

*Essential revisions:*

*1) Because the paper is long and so verbose, the main point of this paper is hard to understand. All of the main texts and figures should be more concise. Many figures are of technical nature and should perhaps be shown in the supplement. In the present manuscript version, the important messages drown in a wealth of material of more technical nature.*

We have rewritten several sentences and sections, moved six technical figures to supplemental materials, removed an additional three technical figures and one performance metric (see below). We have removed repetitive text from the main part of the manuscript. In addition, we have created two additional figures (one main figure, comparing biomarker lists of each model; one supplemental figure comparing performance of models given differing number of input samples), put more emphasis on some of the conceptual issues and provided a more extended biological-molecular interpretation of the main findings. This should better convey the central message of this manuscript.

*2) The SCN phase is not the same as the melatonin phase, the authors describe "predicting the circadian phase of the SCN" throughout the manuscript. As the authors described in Discussion, actual error in SCN phase measurement by melatonin is not known. The authors should use "melatonin phase" instead of "SCN phase". In addition, knowing the melatonin phase is not a panacea for delivering drugs at appropriate times for all diseases (for example, knowing the phase of the liver for a liver-specific disease may be more relevant than knowing the melatonin phase). The authors need to add discussion for this point.*

A) We agree that SCN phase is not melatonin phase although in human clinical circadian and sleep research it is used as the gold standard proxy for SCN phase, which is relevant of chronotherapy (by light or melatonin) aimed at correcting SCN phase. In most previous attempts to develop markers for circadian phase it was often not specified which circadian phase was assessed and conditions of desynchrony were not considered. We attempted to improve on these previous publications by specifying which circadian phase we wanted to predict and by introducing a condition in which circadian rhythms in various organs may desynchronise. In response to this comment we have changed the title to “Blood transcriptome based biomarkers for human circadian phase” and replaced SCN phase with melatonin phase throughout the text where appropriate.

B) We fully agree and recognise that knowing melatonin phase may not be relevant for e.g. liver phase. Indeed, under conditions of desynchrony between SCN phase and rest-activity and fasting feeding cycles or in disease liver phase may not be related to either SCN phase or melatonin phase. In response to the reviewer’s comment we have extended the Discussion section in which we address this issue and also point out that the blood transcriptome may be used to develop phase markers for other organs.

*3) The authors should make the code available (e.g. an R package, e.g. ZeitZeiger, or Python or whatever). In addition to making the code available, a more detailed description of PLSR should be included.*

A) All code used to build and test the models, calculate the given statistics and generate the provided figures are now provided, along with a description on their usage, will be publicly available on our server at: http://sleep-sysbio.fhms.surrey.ac.uk/PLSR_16/ upon acceptance of this manuscript. This is now pointed out on in the Methods section.

B) We have expanded the description of our PLSR approach in the Methods section.

*4) The authors did a good job of pointing to the primary data. However, they should also document the code/scripts used to run the Molecular timetable method, ZeitZeiger, and PLSR so that others can replicate this analysis.*

All code/script will be publicly available and annotated so that others can replicate the analyses.

*5) What influence does the size of the training set have on PLSR's accuracy?*

We have created a training set and validation set by randomly dividing the available data in two equally sized sets. Our training is cross-validated meaning that the size of the training set should not influence i.e. is not over-fitted to the data. However, like any machine learning, we are restricted by the availability of data, the cohort we have collected from. If we test this ‘theory of influence’ and we selected only 10% of samples to form our training set and get a different answer, does this mean that PLSR has failed? No. It means that the data we have selected is not representative of the overall cohort. What we can assess is the robustness/sensitivity of the method to sample size. We have addressed this by investigating the effect of a reduction in the training set size by reducing the training set to have 50%, 60%, 70%, 80% and 90% of the participants in the complete training set. As shown in Figure 6, we see no significant effect on the performance of our model when applied to the data in our independent validation set. Thus, we believe our model to be robust and applicable to the wider population.

Author response image 1.**DOI:**
http://dx.doi.org/10.7554/eLife.20214.024

*6) Can PLSR be used to rule out rhythmicity (ala Molecular timetable and ZeitZeiger)?*

No, PLSR cannot be used to rule out rhythmicity. Our aim was to predict circadian phase. In fact, rhythmicity of a transcript under all conditions is not an a priori requirement for the transcript to be included in the predictor set. In other words, the selection method is unbiased with respect to rhythmicity. This is important because assessment of rhythmicity depends on criteria which are somewhat arbitrary. We have added a few sentences in the Discussion to emphasize this point. We also point out that in one of the conditions, rhythmicity was very much disrupted, and the biomarkers we identify for predicting melatonin phase are therefore robust against this disruption in rhythmicity.

*7) The Molecular timetable method has also been used with metabolite data. Does PLSR work with metabolite data as well?*

PLSR is a generic method and widely used to derive predictors for example in chemometrics. There is no reason why ‘metabolomic’ results couldn’t be used as input. Unfortunately, there are currently not enough publicly available metabolite time series with matching melatonin data for our current purpose (i.e. developing a predictor for human melatonin phase which is robust against conditions of sleep deprivation and desynchrony). We have added metabolomics to the sentence describing the other potential applications of this method.

*8) From Table 1 and Figure 4, PLSR under-estimates melatonin phase (negative error value). Is this correct?*

A negative average error means that the predicted phase is on average earlier than the observed phase, and yes this is possible. Please note that the average error (or bias) is not meaningful as a measure of the accuracy of a method. It can easily be corrected for by adding a constant. The true measure of performance is the standard deviation which represent errors which cannot be corrected.

*9) The Agostinelli et al. paper is cited. BIO_CLOCK should be added to the algorithms benchmarked here.*

The tool for implementing BIO_CLOCK is not (yet) available [and according to the website the models are being retrained].

*10) This model improves upon previous models, but what advantage does this improved model have over previous models? Or, what advantage does this model have over just measuring melatonin? For example, what disease would benefit from having a model that has 20% error compared to 10% error? Of course, having a more accurate model is better, but it's not clear what problems or challenges that this new model will solve. Or what issues could be addressed that were not addressed by previous models?*

A) PLSR is more accurate than other approaches and as pointed out in the comments above this is an obvious advantage.

B) As we pointed out in the Introduction, to derive melatonin phase from melatonin measurements requires a long time series of blood samples(i.e. a 24 h time series if no a priori knowledge about phase is available, as is the case in, for example, blind individuals). Our method only requires one or two blood samples and this is obviously an advantage. We have reemphasized this point in the Discussion.

C) There are two fundamental issues that we wanted to address

i) Can we apply an unbiased method (i.e. a method that makes no assumptions about rhythmicity of the predictor) to derive a predictor of circadian phase? All previous methods start from the assumption that the predictors must be rhythmic or are clock genes.

ii) Can we develop a method that can predict circadian melatonin phase even if there is no sleep-wake cycle or if the sleep-wake cycle is desynchronised from the melatonin rhythm? Previous models have not conducted this essential test (To predict circadian phase when the sleep-wake cycle is synchronised is easy; just measure sleep-timing and it will predict circadian phase quite well).

iii) Thus by combining 1 and 2 and comparing this to other approaches we believe that we have addressed issues that are fundamental in this research area.

We have added sentences to the Discussion to better explain this.

*11) The SD of the prediction error is on the order of 2-3 hrs. Is this likely to be enough for the method to be practically useful in e.g. clinical applications?*

2-3 hours is better than 3-4 hours. Furthermore, our aim was to also identify the limitations of prediction methods if tested under relevant conditions (i.e. desynchrony). Finally, yes 2-3 hours may be good enough for certain applications, such as the determination of the timing of melatonin administration to efficiently synchronise blind individuals. If this accuracy is not good enough then this method can certainly be used to determine the clock time window during which for example a dim light melatonin onset (which last 6-8 hours) should be scheduled. We have addressed this in the Discussion.

*12) Authors should explain why they think only PER1 was identified in both molecular timetable methods (subsection “Molecular Timetable and Zeitzeiger methods”, second paragraph).*

Molecular time table methods as applied in this manuscript require a transcript to be rhythmic in all conditions. PER1 is robustly rhythmic, i.e. also in the absence of sleep and during desynchrony and therefore ‘survives’. Other genes are less robustly rhythmic across the four conditions we have used to identify putative phase markers.

As an example, there are 12 Lech et al. genes. Which are targeted by 153 probes on our microarray. The probes include ‘custom’ probes of ours that potentially target different splice variants of ‘classical’ clock genes (see [Supplementary-material SD4-data]) and Agilent standard probes. Of the 29 Agilent standard probes targeting the genes of Lech et al., a total of 26 probes passed our probe quality control assessments. Given that some genes are targeted by more than one probe, for each gene, the standard probe with the highest correlations was used to represent each gene.

Author response image 2.**DOI:**
http://dx.doi.org/10.7554/eLife.20214.025

As can be seen that for the 12 Lech et al. genes:

ARNTL, CAPRIN1, CLOCK, PER3, ROCK2 do not reach the Pearson correlation cut-off of 0.3 between the templates and the data in any of the four datasets.

SIRT1, STAT3, THRA and TRIB1 only reach the 0.3 cut-off in the Sleep in phase with melatonin dataset.

HSPA1A reaches the 0.3 cut-off in the Sleep out of phase with melatonin and Total sleep deprivation with no prior sleep debt datasets.

MKNK2 reaches the 0.3 cut-off in all datasets except for the Total Sleep deprivation with prior sleep debt dataset.

PER1 reaches the 0.3 cut-off in all four datasets.

*13) Authors should explain why they think those particular 4 genes were identified as circadian in all four conditions (subsection “Molecular Timetable and Zeitzeiger methods”, second paragraph). Because other hnRNPs can directly regulate AANAT, the rate-limiting step in melatonin biosynthesis, is it possible that hnRNPDL is also directly regulating AANAT (as opposed to, or in addition to, NRF as the authors suggest in their model)? And could this regulation occur via locally produced melatonin from peripheral blood mononuclear leucocytes?*

The reference here to four of the Timetable genes that were classified as being rhythmic in our previous published analyses is unclear and detracts from our main results. We have removed this from the Results and have instead included reference to HNRPDL and melatonin in the Discussion.

*14) In Figure 2, the cutoff correlation value of 0.3 is slightly arbitrary because there is so much variation in model performance with slightly different correlation values. For example, a cut-off correlation value of 0.4 performs almost as well as 0.3. Authors should discuss about this point.*

We have now included a more explicit statement in the Methods section to describe how we selected the threshold of 0.3, based on R^2^ values of cross-validated models constructed using varying thresholds of correlation (as indicated, and pointed to, in (now) Figure 2—figure supplement 1).